# CDFlow: Building Invertible Layers with Circulant and Diagonal Matrices

**Xuchen Feng**
School of Integrated Circuits
Sun Yat-sen University
fengxch@mail2.sysu.edu.cn

**Siyu Liao**[*]
School of Integrated Circuits
Sun Yat-sen University
liaosy36@mail.sysu.edu.cn

## Abstract

Normalizing flows are deep generative models that achieve efficient likelihood estimation and sampling through invertible transformations. A key challenge is designing linear layers that enhance expressiveness while enabling efficient computation of the Jacobian determinant and inverse. In this work, we introduce a novel invertible linear layer based on the product of circulant and diagonal matrices. This decomposition provides a parameter- and computation-efficient formulation, reducing the parameter complexity from $\mathcal{O}(n^2)$ to $\mathcal{O}(mn)$ by using $m$ diagonal matrices together with $m-1$ circulant matrices, while approximating arbitrary linear transformations. Furthermore, leveraging the Fast Fourier Transform (FFT), our method reduces the time complexity of matrix inversion from $\mathcal{O}(n^3)$ to $\mathcal{O}(mn \log n)$ and matrix log-determinant from $\mathcal{O}(n^3)$ to $\mathcal{O}(mn)$, where $n$ is the input dimension. Building upon this, we introduce a novel normalizing flow model called Circulant-Diagonal Flow (CDFlow). Empirical results demonstrate that CDFlow excels in density estimation for natural image datasets and effectively models data with inherent periodicity. In terms of computational efficiency, our method speeds up the matrix inverse and log-determinant computations by $1.17\times$ and $4.31\times$, respectively, compared to the general dense matrix, when the number of channels is set to 96.

## 1 Introduction

Deep generative models have gained significant attention due to their potential applications across various domains, including natural language processing [Dong et al., 2024], image restoration and generation [Yao et al., 2023], robot learning [Chao et al., 2024], and autonomous driving [Guan et al., 2024]. Among these models, normalizing flows hold a distinctive position because of their ability to provide accurate likelihood estimation and efficient sampling.

Flow-based models enable the probabilistic modeling of complex distributions by transforming them into simpler ones through a series of invertible mappings. Their uniqueness lies in the strict requirements for the construction of these functions: on one hand, the functions must be invertible; on the other hand, the determinant of the corresponding Jacobian matrix should be easy to compute. However, these constraints may limit the expressive power of the models.

To address this challenge, NICE [Dinh et al., 2014] introduced additive coupling layers, which were later extended by RealNVP [Dinh et al., 2016] to affine coupling layers. These coupling layers have since become a fundamental component of most flow-based models. However, the design of these non-linear layers, which only transform a subset of variables, requires coordination with linear layers to effectively capture dependencies across arbitrary dimensions. As a result, recent

---

[*]Corresponding author.

39th Conference on Neural Information Processing Systems (NeurIPS 2025).

efforts have focused on developing specialized linear layer designs, ranging from fixed channel-disrupting operations [Dinh et al., 2014, 2016] to $1 \times 1$ convolutions [Kingma and Dhariwal, 2018] and even $d \times d$ convolutions [Hoogeboom et al., 2019] with learnable parameters. However, dense convolutions can result in significant memory overhead as a result of the large weight matrices required for high-dimensional inputs. One approach to address this is the construction of reversible butterfly layers [Meng et al., 2022] using butterfly matrices [Dao et al., 2019], which help maintain the model's expressiveness while reducing the number of parameters.

Structured matrices and Fourier transform acceleration have been demonstrated in numerous studies [Cheng et al., 2015, Ding et al., 2017, Karami et al., 2019, Jensen et al., 2021] to be highly effective in reducing the number of parameters and computational complexity of neural networks. Research by [Huhtanen and Perämäki, 2015] demonstrates that any matrix $\mathbf{M} \in \mathbb{R}^{n \times n}$ can be represented as the alternating product of a cyclic matrix and a diagonal matrix, requiring no more than $2n - 1$ factors. This factorization structure has demonstrated versatility and effectiveness in domains such as deep neural networks [Moczulski et al., 2015, Araujo et al., 2019] and parameter-efficient fine-tuning [Ding et al., 2025]. However, these studies have not focused on model reversibility.

Inspired by previous work, we propose a novel normalizing flow model called Circulant-Diagonal Flow (CDFlow), which is built on invertible linear layers constructed using circulant and diagonal matrices. This construction preserves the expressive power of the model while significantly reducing storage demands, as only the diagonal and circulant elements need to be stored instead of the full $n \times n$ weight matrix. In addition, the distinct properties of circulant and diagonal matrices enable the use of the Fast Fourier Transform (FFT) to enhance computation efficiency [Ding et al., 2025]. This facilitates faster calculations for both the determinant of Jacobian matrices and the inversion of weight matrices. Overall, the main contributions of our work can be summarized as following:

- We propose CDFlow, a new class of flow-based generative models. The linear layer in CDFlow is designed to have a storage complexity that scales linearly with the input dimension, as it only requires storing a collection of circulant and diagonal vectors — namely, the diagonal elements of diagonal matrices and the eigenvalue vectors of circulant matrices. In practice, the linear layer weights can be constructed using a small number of circulant and diagonal matrices, enabling efficient transformations across both spatial and channel dimensions.

- We leverage the unique properties of circulant and diagonal matrices to significantly reduce computational complexity. A key property is that the determinant of any diagonalizable matrix equals the product of its eigenvalues. Accordingly, by performing an $\mathcal{O}(n \log n)$ FFT once during initialization to precompute the eigenvalues of circulant matrices, we enable $\mathcal{O}(n)$ log-determinant evaluation at runtime. Additionally, the complexity of matrix inversion is reduced to $\mathcal{O}(n \log n)$.

- We demonstrate through image generation experiments that our CDFlow performs effectively while also offering superior computational efficiency in terms of both log-determinant computation and matrix inverse.

## 2 Background

### 2.1 Flow-based Models

The normalizing flow model maps the complex data distribution $p_X(x)$ to a known simple probability distribution $p_Z(z)$ via an invertible transformation $f_\theta : x \to z$ for data generation with $\theta$ being the model parameter. The log-likelihood $\log p_\theta(x)$ can be obtained according to the change of variables formula:

$$\begin{aligned}
\log p_\theta(x) &= \log p_Z(z) + \log |\det(\frac{\partial f_\theta}{\partial x})| \\
&= \log p_Z(z) + \log |\det\left(J_{f_\theta}(f_\theta^{-1}(z))\right)|,
\end{aligned} \quad (1)$$

where $J_{f_\theta}(\cdot)$ calculates the Jacobian matrix of the transformation $f_\theta(x)$, and $\det(\cdot)$ computes the matrix determinant.

It should be noted that $f_\theta$ can be composed of multiple layers in the case of using deep learning models, and each layer needs to be an invertible function to guarantee the invertibility of $f_\theta$:

$$\begin{aligned} f_\theta(x) &= f_L \circ f_{L-1} \circ \cdots \circ f_1(x), \\ f_\theta^{-1}(z) &= f_1^{-1} \circ f_2^{-1} \circ \cdots \circ f_L^{-1}(z), \end{aligned} \tag{2}$$

where $\circ$ is for function composition and there are $L$ layers in the model. According to the chain rule in differentiability, the determinant of $f_\theta$ is now transformed into layer-wise computation:

$$\log \left| \frac{\partial f_\theta}{\partial x} \right| = \log \left| \frac{\partial f_1}{\partial x} \right| + \sum_{i=2}^{L} \log \left| \frac{\partial f_i}{\partial f_{i-1}} \right|. \tag{3}$$

Therefore, it is important to have an efficient log-determinant computation when training normalizing flow models. When $f_i$ is a linear layer with weight matrix $\mathbf{W}$, its Jabobian matrix is equivalent to the weight matrix:

$$\log |\det \frac{\partial f_i}{\partial f_{i-1}}| = \log |\det \mathbf{W}|, \tag{4}$$

where the determinant computation complexity is $\mathcal{O}(n^3)$.

Normalizing flow model can perform data generation using $f_\theta^{-1}$ by feeding with samples $z$ from $p_Z(z)$. Therefore, it is important to have an efficient $f_\theta^{-1}$ to achieve fast data generation. In particular, when $f_i$ is a linear layer, its inverse is equivalent to the linear transformation built on top of the inverse of weight matrix $\mathbf{W}$:

$$f_i^{-1}(x) = \mathbf{W}^{-1} \times x. \tag{5}$$

In general, the matrix inverse can take $\mathcal{O}(n^3)$ computation complexity.

Overall, log-determinant computation needs to be fast for training normalizing flow models, while data generation requires fast inverse computation.

**Nonlinear Coupling Layer.** The coupling layer [Dinh et al., 2014, 2016] is a simple and effective invertible transformation, where the Jacobi matrix is a triangular matrix. It divides the input into two parts $x_a$ and $x_b$, then transforms $x_b$ using a scaling parameter $s_\theta$ and a translation parameter $b_\theta$ that can be a nonlinear function of $x_a$, while $z_a$ is directly equal to $x_a$, and the final output concatenates the two parts together:

$$z_a = x_a, \tag{6}$$
$$z_b = s_\theta \odot x_b + b_\theta, \tag{7}$$
$$z = \mathrm{concat}(z_a, z_b). \tag{8}$$

Coupling layers remain a crucial component in most flow-based models [Kingma and Dhariwal, 2018, Hoogeboom et al., 2019, Meng et al., 2022]. However, since coupling layers only transform a subset of the variables, they must be combined with linear layers to facilitate information fusion across channels. As a result, recent research has increasingly focused on the design of linear layers to enhance model performance.

**Invertible Linear Layer.** A well-designed linear layer can significantly enhance both the coupling layer's ability to learn the target distribution and the overall expressive power of the model. Furthermore, the computational cost associated with matrix inversion and determinant computation serves as an important indicator of the efficiency of linear layer design.

Glow [Kingma and Dhariwal, 2018] pioneered the use of $1 \times 1$ convolution with learnable parameters as a linear layer in flow-based models. This method is more effective at capturing correlations between variables compared to simpler channel shuffling operations [Dinh et al., 2014, 2016]. Additionally, Glow introduces a weight design based on LU decomposition, reducing the Jacobian determinant computation from $\mathcal{O}(n^3)$ to $\mathcal{O}(n)$.

Similarly, [Hoogeboom et al., 2019] extended convolution to $d \times d$ and proposed periodic convolution, leveraging the Fast Fourier Transform (FFT) to accelerate the convolution process. However, the periodic convolution method involves generalized determinant computation, leading to a complexity of $\mathcal{O}(n^3)$. In contrast, our proposed method not only accelerates the convolution process using FFT, but also reduces the determinant computation complexity to $\mathcal{O}(n)$.

Despite numerous works exploiting special matrix properties to enhance model expressiveness, most focus on improving the efficiency of the Jacobian determinant computation, with limited optimization of the matrix inversion process during the sampling phase of the flow model. In these methods, including the aforementioned works [Kingma and Dhariwal, 2018, Hoogeboom et al., 2019, Meng et al., 2022], the computational complexity for the inverse sampling operation is typically $\mathcal{O}(n^2)$. The Woodbury Transformation [Lu and Huang, 2020] achieves a theoretical complexity of $\mathcal{O}(dn)$, where $d$ is the dimensionality of smaller matrices (e.g., 8 or 16). Although this is better in theory than the $\mathcal{O}(n \log n)$ complexity of our method, in practice Woodbury requires multiple input transformations, which makes its actual efficiency inferior to ours.

## 2.2 Weight Matrix Construction and Fast Multiplication

A general matrix $\mathbf{M} \in \mathbb{R}^{n \times n}$ can be factorized into the alternating product of circulant and diagonal matrices, with the total number of matrices not exceeding $2n - 1$ [Huhtanen and Perämäki, 2015]. Specifically, it can be written as:

$$\mathbf{M} = \mathbf{D}_1 \mathbf{C}_2 \mathbf{D}_3 \ldots \mathbf{D}_{2n-3} \mathbf{C}_{2n-2} \mathbf{D}_{2n-1}, \tag{9}$$

where $\mathbf{D}_{2j-1}$ and $\mathbf{C}_{2j}$ are diagonal and circulant matrix for $j = 1, 2, \ldots, n$. This decomposition can theoretically approximate any dense matrix, and the approximation error can be controlled by adjusting the number of factors. [Ding et al., 2025] have successfully applied such representation to fine-tuning large language models.

Denote the input vector as $\mathbf{x} \in \mathbb{R}^n$. The weight matrix $\mathbf{W} \in \mathbb{R}^{n \times n}$ can be represented by $m$ diagonal matrices and $m - 1$ circulant matrices, thereby $m \leq n$. For each $j \in \{1, 2, \ldots, m\}$, the $j$-th diagonal matrix is defined by the vector $\mathbf{d}_{2j-1} \in \mathbb{R}^{n \times 1}$, and the $j$-th circulant matrix is defined by the vector $\mathbf{c}_{2j} \in \mathbb{R}^{n \times 1}$. Thus, the model weight matrix can be represented as:

$$\mathbf{W} = \mathbf{D}_1 \mathbf{C}_2 \mathbf{D}_3 \ldots \mathbf{D}_{2m-3} \mathbf{C}_{2m-2} \mathbf{D}_{2m-1}$$
$$= \operatorname{diag}(\mathbf{d}_1) \times \operatorname{circ}(\mathbf{c}_2) \times \cdots \times \operatorname{diag}(\mathbf{d}_{2m-1}), \tag{10}$$

where $\operatorname{diag}(\cdot)$ and $\operatorname{circ}(\cdot)$ represent the construction of diagonal and circulant matrices, respectively. It should also be noted that circulant matrix can be factorized into:

$$\operatorname{circ}(\mathbf{c}_{2j}) = \mathbf{F}^{-1} \times \operatorname{diag}(\hat{\mathbf{c}}_{2j}) \times \mathbf{F}, \quad \hat{\mathbf{c}}_{2j} = \mathbf{F} \times \mathbf{c}_{2j}, \tag{11}$$

where $\mathbf{F}$ is the order $n$ discrete Fourier transform matrix. In practice, it is better to implement with the frequency domain parameter $\hat{\mathbf{c}}_{2j}$ as weight parameter rather than real valued $\mathbf{c}_{2j}$ to save computation costs in matrix vector product, logarithm determinant, and matrix inversion.

# 3 Invertible Linear Layer Design

In this section, we introduce our invertible linear layer that consist of circulant and diagonal matrices. We discuss the fast algorithms for the matrix vector product, logarithm of matrix determinant and matrix inversion. Additionally, we also analyze the theoretical computation cost associated with these operations.

## 3.1 Matrix Log-determinant

According to Eq. 10, The weight matrix is constructed as an alternating product of diagonal and circulant matrices. The determinant of this matrix can be efficiently computed by leveraging the special properties of these matrices. Recall that determinant of matrix product is equal to product of matrix determinant:

$$\det(\mathbf{W}) = \prod_{j=1}^{m} \det(\mathbf{D}_{2j-1}) \times \prod_{j=1}^{m-1} \det(\mathbf{C}_{2j}), \tag{12}$$

$$\det(\mathbf{D}_{2j-1}) = \det(\operatorname{diag}(\mathbf{d}_{2j-1})) = \prod_{i=1}^{n} d_{2j-1}^i, \tag{13}$$

$$\det(\mathbf{C}_{2j}) = \det(\mathbf{F}^{-1}) \det(\operatorname{diag}(\hat{\mathbf{c}}_{2j})) \det(\mathbf{F}) = \prod_{i=1}^{n} \hat{c}_{2j}^i. \tag{14}$$

Table 1: Comparisons of computational complexities and parameter counts, where $n$ is the number of input channels. In practice, $d$ is the dimensionality of the smaller matrices used in the Woodbury transformation (typically $d = 8$ or 16), $L$ is the number of butterfly layers, and $m$ is generally set to 2 for our method. Here $h$ and $w$ denote the spatial dimensions of the feature maps.

| Method | Logdet | Inverse | Params |
|---|---|---|---|
| $1 \times 1$ Conv [Kingma and Dhariwal, 2018] | $\mathcal{O}(n^3)$ | $\mathcal{O}(n^3)$ | $\mathcal{O}(n^2)$ |
| $1 \times 1$ Conv (LU) [Kingma and Dhariwal, 2018] | $\mathcal{O}(n)$ | $\mathcal{O}(n^2)$ | $\mathcal{O}(n^2)$ |
| Emerging [Hoogeboom et al., 2019] | $\mathcal{O}(n)$ | $\mathcal{O}(n^2)$ | $\mathcal{O}(n^2)$ |
| Periodic [Hoogeboom et al., 2019] | $\mathcal{O}(n^3)$ | $\mathcal{O}(n^2)$ | $\mathcal{O}(n^2)$ |
| Woodbury [Lu and Huang, 2020] | $\mathcal{O}(nd)$ | $\mathcal{O}(nd)$ | $\mathcal{O}(d(n + h \cdot w))$ |
| ME-Woodbury [Lu and Huang, 2020] | $\mathcal{O}(nd)$ | $\mathcal{O}(nd)$ | $\mathcal{O}(d(n + h + w))$ |
| ButterflyFlow [Meng et al., 2022] | $\mathcal{O}(n)$ | $\mathcal{O}(n^2)$ | $\mathcal{O}(nL)$ |
| CDFlow (Ours) | $\mathcal{O}(mn)$ | $\mathcal{O}(mn \log n)$ | $\mathcal{O}(mn)$ |

To compute the logarithm of the matrix determinant of $\mathbf{W}$, we can express into the sum of each matrix determinant:

$$\log |\det(\mathbf{W})| = \sum_{j=1}^{m} \sum_{i=1}^{n} \log |\hat{c}_{2j}^i| + \sum_{j=1}^{m-1} \sum_{i=1}^{n} \log |d_{2j-1}^i|. \tag{15}$$

Therefore, the overall log-determinant computation can be finished by simple summation with a complexity of $\mathcal{O}(mn)$.

## 3.2 Matrix Inverse

The inversion of the weight matrix for the data sampling process can be efficiently implemented by leveraging the decomposition of the weight matrix into alternating diagonal and circulant matrices. This section introduces the inverse calculation process, including its objective, step-by-step procedure, and computational complexity. The objective of the matrix inversion is to recover the input vector $\mathbf{x}$ from the output $\mathbf{y}$, given the weight matrix $\mathbf{W}$. This is achieved by solving the equation:

$$\mathbf{x} = \mathbf{W}^{-1} \times \mathbf{y}, \tag{16}$$

where the inverse of $\mathbf{W}$ is represented as:

$$\mathbf{W}^{-1} = \mathbf{D}_{2m-1}^{-1} \times \cdots \times \mathbf{C}_2^{-1} \mathbf{D}_1^{-1}. \tag{17}$$

**Diagonal Matrices.** For each diagonal matrix $\mathbf{D}_{2j-1}$, its inverse is obtained by taking the reciprocal of its diagonal elements.

$$\mathbf{D}_{2j-1}^{-1} = \text{diag}(1/d_{2j-1}^1, 1/d_{2j-1}^2, \ldots, 1/d_{2j-1}^n), \tag{18}$$

which has a computation complexity of $\mathcal{O}(n)$.

**Circulant Matrices.** For each circulant matrix $\mathbf{C}_j$, its inverse is similar to a diagonal matrix according to Eq. (11):

$$\begin{aligned} \mathbf{C}_{2j}^{-1} &= \mathbf{F}^{-1} \times \text{diag}^{-1}(\hat{\mathbf{c}}_{2j}) \times \mathbf{F} \\ &= \mathbf{F}^{-1} \times \text{diag}(1/\hat{c}_{2j}^1, \ldots, 1/\hat{c}_{2j}^n) \times \mathbf{F}. \end{aligned} \tag{19}$$

It can be seen that matrix inversion $\mathbf{W}^{-1}$ can be immediately determined given $\mathbf{d}_{2j-1}$ and $\mathbf{c}_{2j}$. Therefore, the complexity of Eq. (16) is equivalent to matrix vector product complexity, i.e., $\mathcal{O}(mn \log n)$.

## 3.3 Generative Modeling with CDFlow

We present our CDFlow model that consists of $L$ blocks, where each block is formed by stacking $K$ layers of flow modules. Each flow module consists of the ActNorm layer, the invertible convolutional layer, and the coupling layer, working together to transform the input data. The multi-scale structure helps the model learn complex relationships within the data by processing it at different levels of abstraction. Figure 1 illustrates how the flow module is constructed for each layer and how the entire model is repeated and integrated to learn and generate the target data.

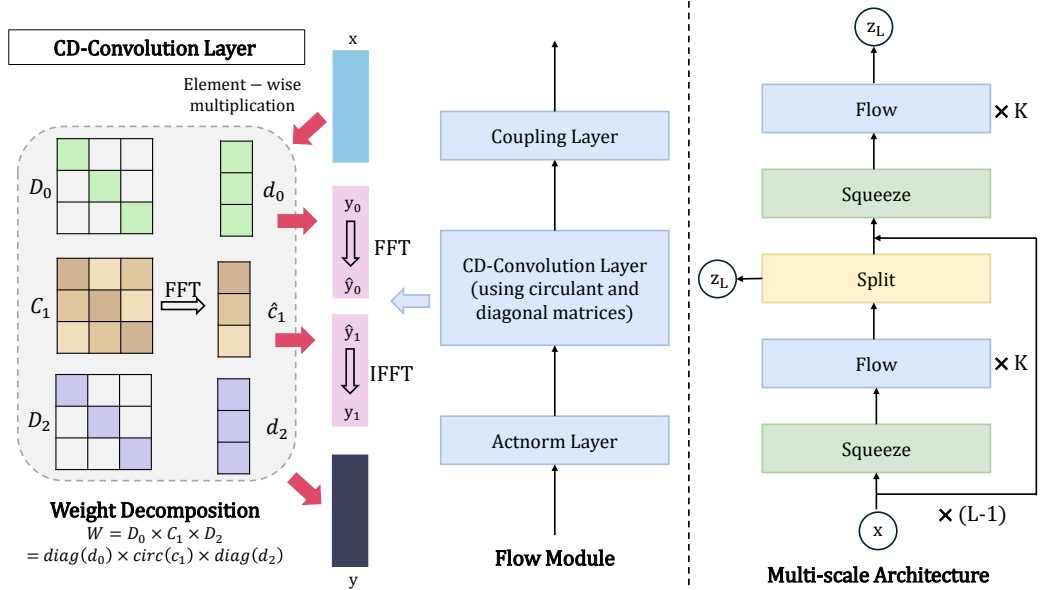

Figure 1: Architecture of the CDFlow Model. The left portion shows a detailed schematic of the proposed CD-Convolution layer, which is constructed using circulant and diagonal matrices. The middle portion illustrates the internal structure of a flow module, where the input sequentially passes through an Actnorm layer[Kingma and Dhariwal, 2018], the CD Convolution layer, and a coupling layer[Dinh et al., 2016]. The right portion presents the overall CDFlow model architecture, in which the flow module is repeated K times and integrated into a multi-scale framework[Dinh et al., 2016] to effectively fuse channel-wise information.

**Actnorm Layers.**    The ActNorm layer [Kingma and Dhariwal, 2018] is a type of normalization technique that uses affine transformations to perform class normalization. Unlike traditional methods like BatchNorm [Ioffe and Szegedy, 2015], which rely on the statistics of the entire batch, the ActNorm layer initializes its parameters based on the statistics of the first batch. Once initialized, these parameters are treated as learnable variables. This initialization method is advantageous for small batch sizes, as it avoids the performance degradation that BatchNorm often encounters when dealing with small batches. ActNorm helps in stabilizing the training of normalizing flows and is especially useful for flow-based models that require flexible scaling and translation invariance.

**Invertible Convolutional Layer.**    In CDFlow, the CD-Convolution layer leverages special weight matrices, specifically circulant and diagonal matrices, to enable efficient computation. This design significantly reduces the computational cost of matrix inversion and the computation of the Jacobi determinant, which are essential operations for flow-based models. A current limitation of our approach is its reliance on $1 \times 1$ convolution, and our future work will extend to $d \times d$ convolutions. A detailed explanation of this design can be found in Section 3.

**Coupling Layers.**    The coupling layer [Dinh et al., 2014, 2016] is a core component of normalizing flow models. The coupling layer splits the input into two parts: one part is passed through a neural network to compute a transformation, while the other part is left unchanged. This operation enables the model to learn complex transformations while maintaining a simple and computationally efficient structure. The coupling layer is combined with the reversible convolutional layer to form the flow module, allowing the CDFlow model to capture complex distributions and generate high-quality samples.

**Multi-scale architecture.**    CDFlow incorporates a multi-scale architecture [Dinh et al., 2016], which improves the model's ability to capture complex and high-dimensional data distributions. Within each block, the split and squeeze operations play an important role:

- The **split** operation divides the input tensor along the channel dimension, which allows the model to independently process different features. This enhances the model's expressive power by enabling it to focus on diverse aspects of the data.
- The **squeeze** operation reduces the spatial resolution of the data by downsampling the spatial dimensions. This operation allows the model to focus on higher-level features while also reducing computational cost by focusing on the most important data representations.

## 4 Experiments

In this section, we present the experimental results to evaluate the performance of the proposed method. The experiments are conducted in three main parts:

1. We first evaluate the method on commonly used natural image datasets to demonstrate its general effectiveness under standard conditions.
2. Next, we apply the method to datasets with specialized structures to investigate its performance on data exhibiting periodic patterns.
3. Finally, we conduct a runtime analysis to assess the computational efficiency of the method.

The following subsections present the detailed results and discussions for each of these experiments.

We further investigated the effectiveness of the proposed linear layer within the Flow Matching model, and the results can be found in Appendix C.

### 4.1 Experimental Settings and Model Parameters

**Datasets and Baselines.** To compare our method with previous approaches, we evaluate its performance on the CIFAR-10 [Krizhevsky, 2009] and ImageNet [Deng et al., 2009] datasets, using bits per dimension (BPD) as the evaluation metric. BPD is a commonly used measure for assessing the quality of generative models, indicating the number of bits required to encode each dimension of the data. Metrics such as IS and FID are not used in this work, as they do not generalize well across different datasets and fail to account for overfitting. For the baseline, we select several models related to CDFlow, including MAF [Papamakarios et al., 2017], Real NVP [Dinh et al., 2016], Glow [Kingma and Dhariwal, 2018], Emerging [Hoogeboom et al., 2019], i-ResNet [Behrmann et al., 2019], Woodbury [Lu and Huang, 2020] and ButterflyFlow [Meng et al., 2022].

**Implementation Details.** To facilitate a fair comparison with prior architectures, we adopt consistent backbone settings for flow-based baselines that share a similar multi-scale design, including Glow, Woodbury, Emerging, and ButterflyFlow. On standard images datasets, all of these models are configured with three blocks and 32 flow steps. On the structured dataset, we instead adopt a lighter configuration with two blocks and eight flow steps. For Woodbury, we adhere to the original configuration with $d_s = d_c = 16$, while for ButterflyFlow we use the default setting with butterfly level $= 1$. For our proposed CDFlow, we set $m = 2$, meaning we use two diagonal vectors and a circulant vector.

During training, we apply spectral normalization [Miyato et al., 2018] to the convolutional layers to ensure stability. Thanks to the structure of circulant and diagonal matrices, the maximum singular value of each weight matrix can be efficiently computed and used to rescale the weights. In addition, we employ a channel-aware learning rate scaling for the CD-Convolution parameters, which prevents abrupt updates to the structured matrices. Empirically, this combination further stabilizes the training of CDFlow.

**Hyperparameter Selection** Each linear transform in our method is decomposed into $2m - 1$ matrices (i.e., $m$ diagonal and $m - 1$ circulant). While larger $m$ values increase the approximation power, they also introduce additional parameter and computational costs. The key objective is to balance expressiveness, efficiency, and runtime performance. To this end, we systematically evaluate the effect of varying $m$ on model behavior.

As shown in Figure 2(a), increasing $m$ on CIFAR-10 leads to only marginal improvements in expressiveness, as reflected by BPD values, while significantly increasing the parameter count of the

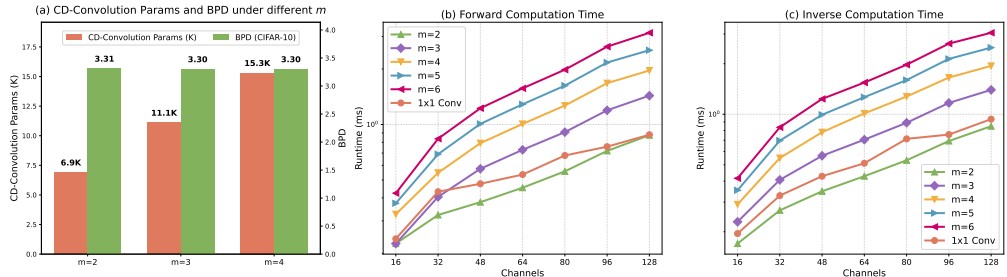

Figure 2: Ablation study on the hyperparameter $m$. Left: parameter counts (K) of the CD-Convolution layers and BPD on CIFAR-10. Middle: forward computation time (ms) across different channel sizes. Right: inverse computation time (ms) across different channel sizes.

Table 2: Density estimation results on CIFAR-10, ImageNet $32 \times 32$, and Galaxy, including the corresponding model sizes (Params) and bits per dimension (BPD). Lower BPD indicates better performance.

| Model | CIFAR-10 | | ImageNet $32 \times 32$ | | Galaxy | |
|---|---|---|---|---|---|---|
| | Params | BPD ($\downarrow$) | Params | BPD ($\downarrow$) | Params | BPD ($\downarrow$) |
| MAF [Papamakarios et al., 2017] | 105.0M | 4.31 | — | — | — | — |
| Real NVP [Dinh et al., 2016] | 6.4M | 3.49 | 46.2M | 4.28 | 6.38M | 2.11 |
| Glow [Kingma and Dhariwal, 2018] | 44.2M | $3.36 \pm 0.002$ | 66.2M | 4.09 | 6.25M | 2.02 |
| Emerging [Hoogeboom et al., 2019] | 46.6M | $3.34 \pm 0.002$ | 44.0M | 4.09 | 6.68M | 1.98 |
| Residual Flows [Chen et al., 2019] | 25.2M | 3.28 | 47.1M | 4.01 | 5.58M | 3.60 |
| i-ResNet [Behrmann et al., 2019] | 25.0M | $3.32 \pm 0.006$ | 29.7M | 4.05 | 6.88M | 2.04 |
| Woodbury [Lu and Huang, 2020] | 45.3M | $3.42 \pm 0.002$ | 45.3M | 4.09 | 6.68M | 2.01 |
| i-DenseNet [Perugachi-Diaz et al., 2021] | 24.9M | 3.25 | 47.0M | 3.98 | 6.07M | 4.06 |
| ButterflyFlow [Meng et al., 2022] | 44.4M | $3.33 \pm 0.003$ | 44.4M | 4.09 | 6.39M | 1.95 |
| CDFlow (Ours) | 44.2M | $3.31 \pm 0.004$ | 44.2M | 4.04 | 6.25M | 1.92 |

CD-Convolution layers. Figures 2(b) and (c) further report the runtime measurements for forward and inverse operations. Although larger $m$ values introduce additional latency, our method remains more efficient than the other methods reported in Section 4.4. In addition, we separately evaluated the log-determinant computation time across channel sizes ranging from 16 to 1024, where the runtime remains consistent across different $m$, with values between 0.095 ms and 0.097 ms.

Based on these results, we adopt $m = 2$ as the default setting for all experiments, including both the main and supplementary studies, unless otherwise specified.

## 4.2 Density Estimation on Standard Images

**Standard Image Datasets.** To enable direct comparison with prior work [Meng et al., 2022], we evaluate our method on the CIFAR-10 [Krizhevsky, 2009] and ImageNet-32×32 [Deng et al., 2009] datasets. For both datasets, training is conducted with unified dequantization and standard data augmentation techniques.

**Results.** Table 2 presents the quantization results of the model on the CIFAR-10 and ImageNet $32 \times 32$ datasets. CDFlow outperforms Glow, Emerging, Woodbury, and ButterflyFlow (all using the same modeling framework) and performs comparably to models like Residual Flows and i-DenseNet. The generated sample results are visualized in Figure 3a and Figure 3b.

## 4.3 Density Estimation on Structured Images

To demonstrate the capability of CDFlow in modeling periodic structures, we apply it to galaxy images [Ackermann et al., 2018], conducting an experiment on this real-world dataset.

**Galaxy Datasets.** When the boundary pixels of an image are relatively uniform or connected, the data can be considered periodic. A typical example of periodic data is galaxy images, which often

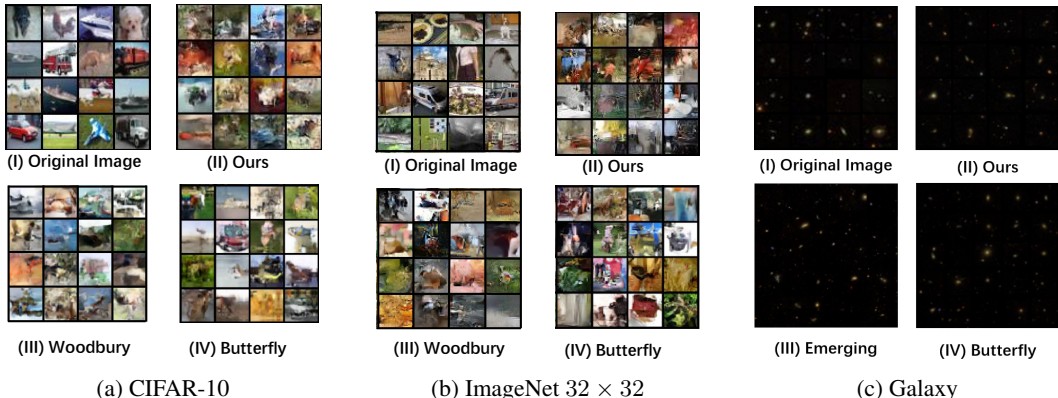

| (I) Original Image | (II) Ours | (I) Original Image | (II) Ours | (I) Original Image | (II) Ours |

| (III) Woodbury | (IV) Butterfly | (III) Woodbury | (IV) Butterfly | (III) Emerging | (IV) Butterfly |

(a) CIFAR-10        (b) ImageNet $32 \times 32$        (c) Galaxy

Figure 3: Comparison between the original image and outputs from different methods. For clearer and larger visualizations, please refer to Appendix D.

feature scattered light sources with predominantly dark boundaries. The Galaxy datasets [Ackermann et al., 2018] is a small classification dataset that includes 3,000 merged galaxies and 5,000 non-merged galaxies in the training and test sets. We train CDFlow on non-merged galaxy images and compare its performance to other models under the same experimental setup.

**Results.** Table 2 also presents the BPD results on the galaxy dataset. It is clear that CDFlow outperforms all other models, achieving a 4.95% improvement in BPD compared to the Glow model. In addition, Figure 3c presents a visual comparison between our method and two strong baselines: Emerging and Butterfly. Experimental results demonstrate that our proposed CDFlow model can well model data with periodic qualities.

## 4.4 Runtime

To demonstrate the effectiveness of our method in reducing computational complexity, we separately evaluated the runtime of the forward operation, log-determinant (logdet) computation, and inverse operation. We compared our method with standard $1 \times 1$ convolution, emerging and periodic convolutions [Hoogeboom et al., 2019], the Woodbury transformation and ME-Woodbury with a fixed latent dimension of $d = 16$ [Lu and Huang, 2020], and the butterfly layer with level 10 [Meng et al., 2022]. For a fair comparison, all methods were implemented in PyTorch, and all experiments were conducted on an NVIDIA A800 GPU.

**Forward Time.** The forward operation refers to the runtime required to apply the tested layer function and compute the log-determinant, i.e., $y = f(x)$ and $\log|\det(\mathbf{W})|$, under gradient computation mode. As shown in Figure 4, although the theoretical complexity of the $1 \times 1$ convolution is $\mathcal{O}(n^3)$, its actual runtime remains relatively stable across varying channel sizes, likely due to GPU parallelism. All baseline methods are slower than the $1 \times 1$ convolution, while our method achieves even faster performance. Notably, although the periodic convolution leverages Fourier transforms, its reliance on two-dimensional FFT introduces substantial memory overhead, leading to sharp runtime increases with more channels. In contrast, our method combines circulant and diagonal matrix constructions to reduce log-determinant complexity, and employs an optimized convolutional implementation to accelerate forward computation, as detailed in Sections 2.2 and 3.1.

In addition, we specifically measured the runtime of the log-determinant computation, which depends solely on the size of the weight matrix, i.e., the number of input channels. Figure 4 illustrates how the runtime of different methods varies with increasing channel dimensions. Our method not only outperforms all baselines but also maintains stable performance without significant growth as the number of channels increases.

**Inverse Time.** The inverse operation refers to the time required to apply the inverse function of the tested layer, i.e., $x = f^{-1}(y)$, under no-gradient mode. We first analyze the impact of channel number on runtime, as shown in Figure 4. The Emerging convolution suffers from poor parallelism, while the

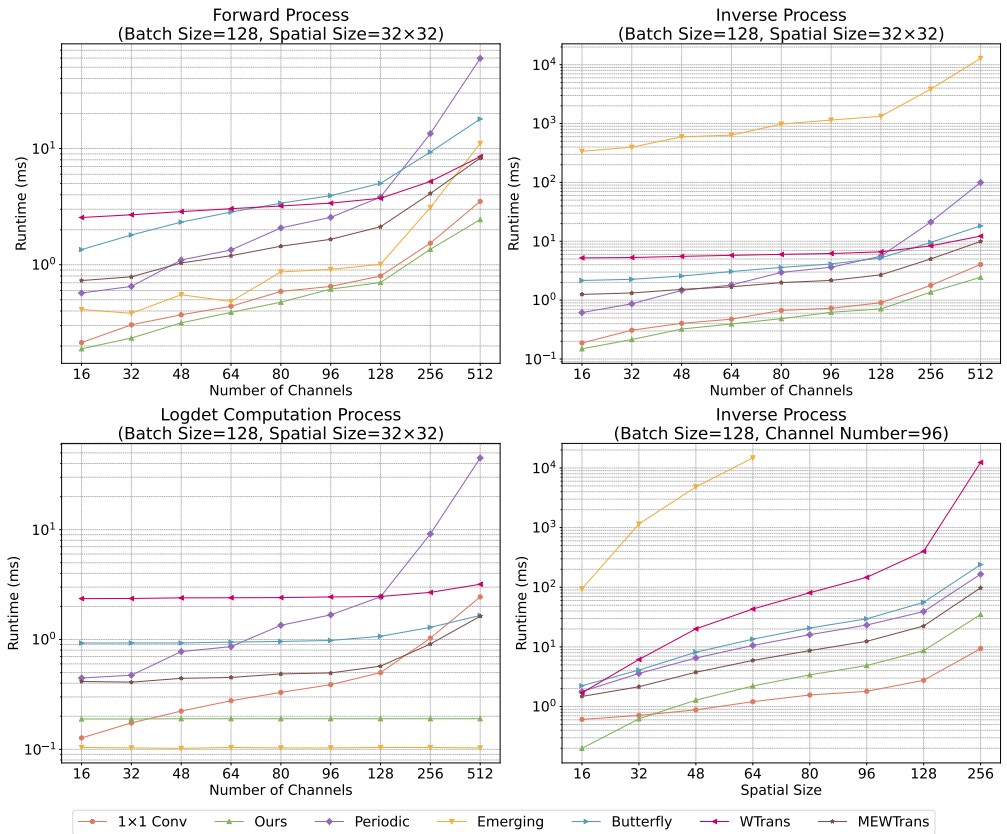

Figure 4: Comparison of runtime (ms) for forward computation, inverse computation, and log-determinant calculation under varying numbers of channels, as well as inverse computation under different spatial sizes. The results represent the average of 100 runs. Our method consistently outperforms baselines, especially in inverse and log-determinant computations. The runtime of the Emerging method becomes excessively long at large spatial sizes and is therefore omitted from the corresponding figure.

periodic convolution relies on two-dimensional Fourier transforms, leading to a significant increase in runtime as the number of channels grows. Other baseline methods perform slightly worse than the $1 \times 1$ convolution, possibly due to the lack of optimized inverse implementations and additional transformation overhead. In contrast, our method reduces the inversion complexity to $\mathcal{O}(n \log n)$, and despite requiring data reshaping, consistently outperforms other baselines, particularly at large channel sizes.

We further evaluate how spatial resolution affects runtime, with the number of channels and batch size fixed. The results show that all methods experience increased runtime with larger spatial sizes. Our method outperforms the $1 \times 1$ convolution at small spatial sizes, and becomes slightly less efficient at larger sizes, since it mainly reduces channel-wise rather than spatial complexity. Nevertheless, our method still clearly outperforms all other baselines except the $1 \times 1$ convolution.

## 5    Conclusion

In this paper, we introduce the CDFlow model, a novel flow-based generative model. Instead of relying on traditional weight matrices, we employ the interleaved multiplication of circulant and diagonal matrices, leveraging the properties of these matrices along with the Fast Fourier Transform (FFT) to efficiently compute the Jacobian determinant and perform the inverse transformation. Our experiments demonstrate that CDFlow excels in density estimation for standard image datasets and performs well on real-world data with periodic properties, outperforming existing baselines.

## Acknowledgments and Disclosure of Funding

This work was financially supported by the National Key R&D Program of China (Grant No. 2024YFA1211400).

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

## A  Impact of Linear Transformation Constraints on Flow-based Model Performance

To further validate the effectiveness of our proposed method, we reproduced the experiment described in Appendix A.5 of Chao et al. [2023] and included an additional variant based on our Diagonal-Circulant Decomposition (DCD) linear layers. The compared linear layer types include **F**-type (full matrix), **L**-type (lower-triangular), **U**-type (upper-triangular), **LU**-type (the product of lower and upper triangular matrices), and our proposed **DCD**-type.

As shown in Fig. 5, the model employing the F-type layers achieved the lowest negative log-likelihood (NLL), consistent with Chao et al. [2023]. Although our DCD-type layer does not surpass the fully learnable F-type, it achieves the second-best performance, outperforming the LU-type design. This result indicates that DCD retains much of the expressiveness of the unconstrained full matrix while maintaining a more efficient and structured parameterization.

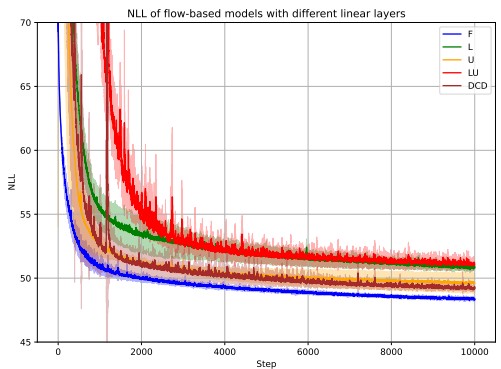

Figure 5: NLL comparison among flow-based models constructed with different linear layer types (F, L, U, LU, and DCD). Our DCD-type layer achieves performance close to the fully learnable F-type and surpasses the LU-type structure.

## B  Runtime

Specifically, our approach requires transforming the input tensor from a four-dimensional structure $[B, C, H, W]$ to a two-dimensional matrix $[B \times H \times W, C]$. While the runtime reported in Section 4.4 omits this transformation, we further measured the total runtime including it to ensure a fair comparison. As illustrated in Figure 6, our method continues to demonstrate strong computational efficiency even after accounting for the transformation overhead. It is worth noting that the computation of the log-determinant is independent of the input tensor, and therefore the reshaping operation has no impact on its runtime.

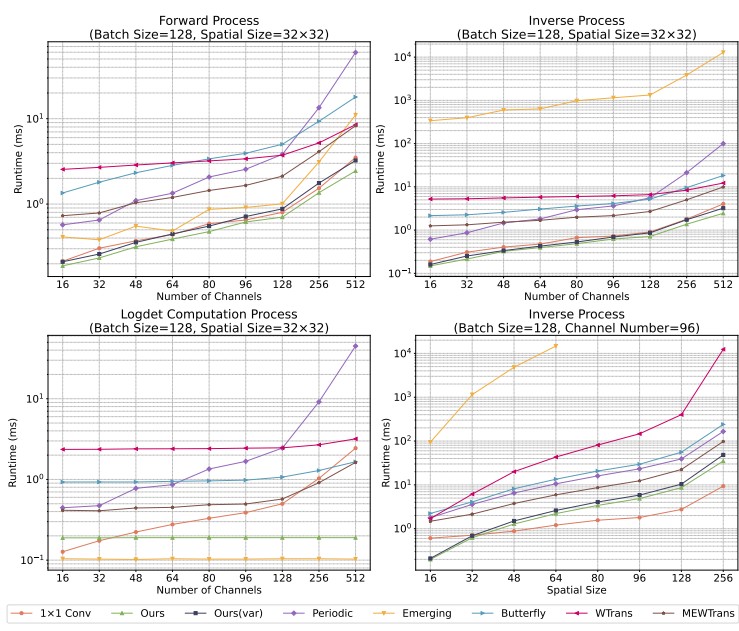

Figure 6: Comparison of runtime (ms) .

# C  Additional Experiments on Flow Matching Models

To further validate the effectiveness of the proposed linear layer, we conducted experiments within the Flow Matching framework Lipman et al. [2022], which employs a 5-layer MLP on 2D toy datasets. To systematically evaluate the method, we performed experiments on three datasets: Chessboard, Moons, and Circles. Within the original framework, we replaced the linear layers of the standard MLP with four structured variants for comparison: EMLP (Emerging Layer) Hoogeboom et al. [2019], WMLP (Woodbury Layer) Lu and Huang [2020], BFMLP (Butterfly Layer) Meng et al. [2022], and CDMLP (Ours).

The experimental results are summarized in Table 3. CDMLP achieves the best or near-best performance across most cases while requiring significantly fewer parameters. For instance, it attains the lowest NLL on the Chessboard and Circles datasets and the best MSE and FID on Moons, outperforming or matching larger models such as MLP and EMLP. In addition, Figures 7–8 further visualize the learned distributions, showing that the model successfully evolves from Gaussian initialization toward the target distributions. These findings not only confirm the modeling capability of the proposed method within the Flow Matching framework but also highlight its applicability and generality across different data structures.

Table 3: Flow Matching Model Results on 2D-toy Datasets. The best results are highlighted in bold.

| Dataset | Model | Params | MSE↓ | FID↓ | NLL↓ |
|---|---|---|---|---|---|
| Chessboard | MLP Lipman et al. [2022] | 791.0K | 3.796 | 0.0062 | 10.6853 |
| | EMLP Hoogeboom et al. [2019] | 1577.5K | 3.797 | 0.0165 | 9.357 |
| | WMLP Lu and Huang [2020] | 40.5K | **3.761** | 0.0042 | 9.0551 |
| | BFMLP Meng et al. [2022] | 26.1K | 3.789 | 0.0125 | 8.2088 |
| | CDMLP (Ours) | **8.45K** | 3.810 | **0.0115** | **7.0425** |
| Moons | MLP Lipman et al. [2022] | 791.0K | 1.854 | 0.0022 | 24.1885 |
| | EMLP Hoogeboom et al. [2019] | 1577.5K | 1.855 | 0.0023 | 36.9462 |
| | WMLP Lu and Huang [2020] | 40.5K | 1.905 | 0.0016 | 47.6328 |
| | BFMLP Meng et al. [2022] | 26.1K | 1.886 | 0.0031 | **17.9168** |
| | CDMLP (Ours) | **8.45K** | **1.842** | **0.0016** | 31.0329 |
| Circles | MLP Lipman et al. [2022] | 791.0K | **2.433** | **0.0019** | 42.4687 |
| | EMLP Hoogeboom et al. [2019] | 1577.5K | 2.517 | 0.0032 | 20.5252 |
| | WMLP Lu and Huang [2020] | 40.5K | 2.547 | 0.0021 | 26.6795 |
| | BFMLP Meng et al. [2022] | 26.1K | 2.470 | 0.0032 | 21.6716 |
| | CDMLP (Ours) | **8.45K** | 2.482 | 0.0069 | **15.4922** |

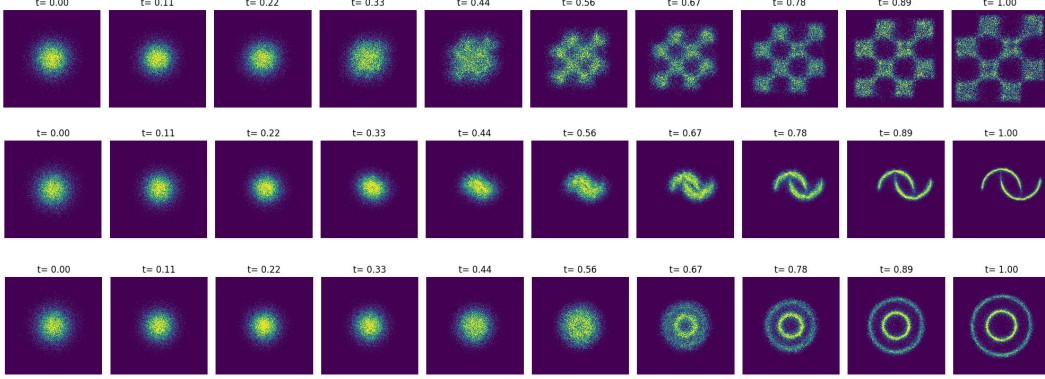

Figure 7: Time evolution of the learned distribution on 2D toy datasets (Chessboard, Moons, Circles).

# D  Visualization of Generation Results on Image Datasets

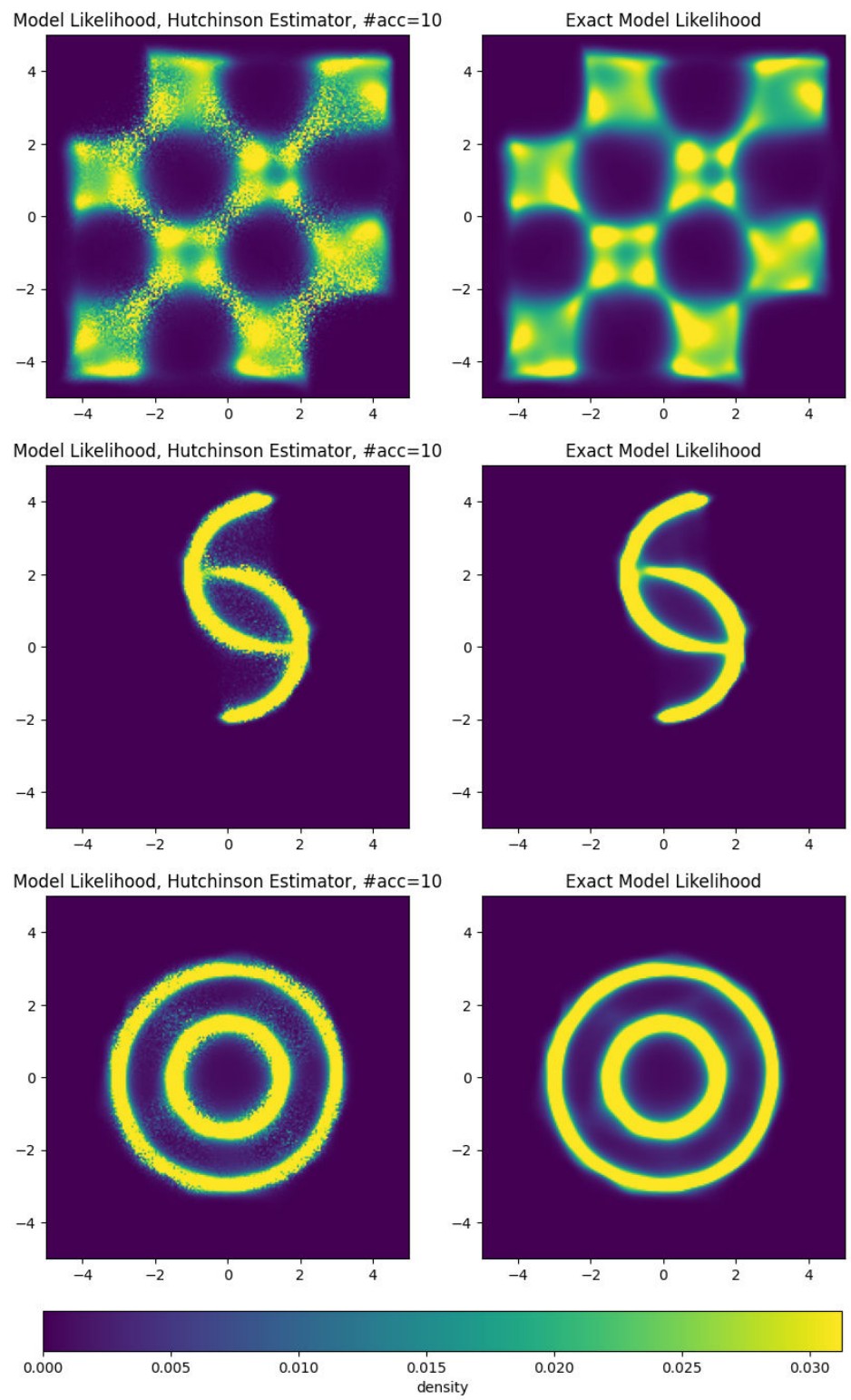

Figure 8: Visualization of model likelihood (left: Hutchinson estimator, right: exact) on 2D toy datasets (Chessboard, Moons, Circles).

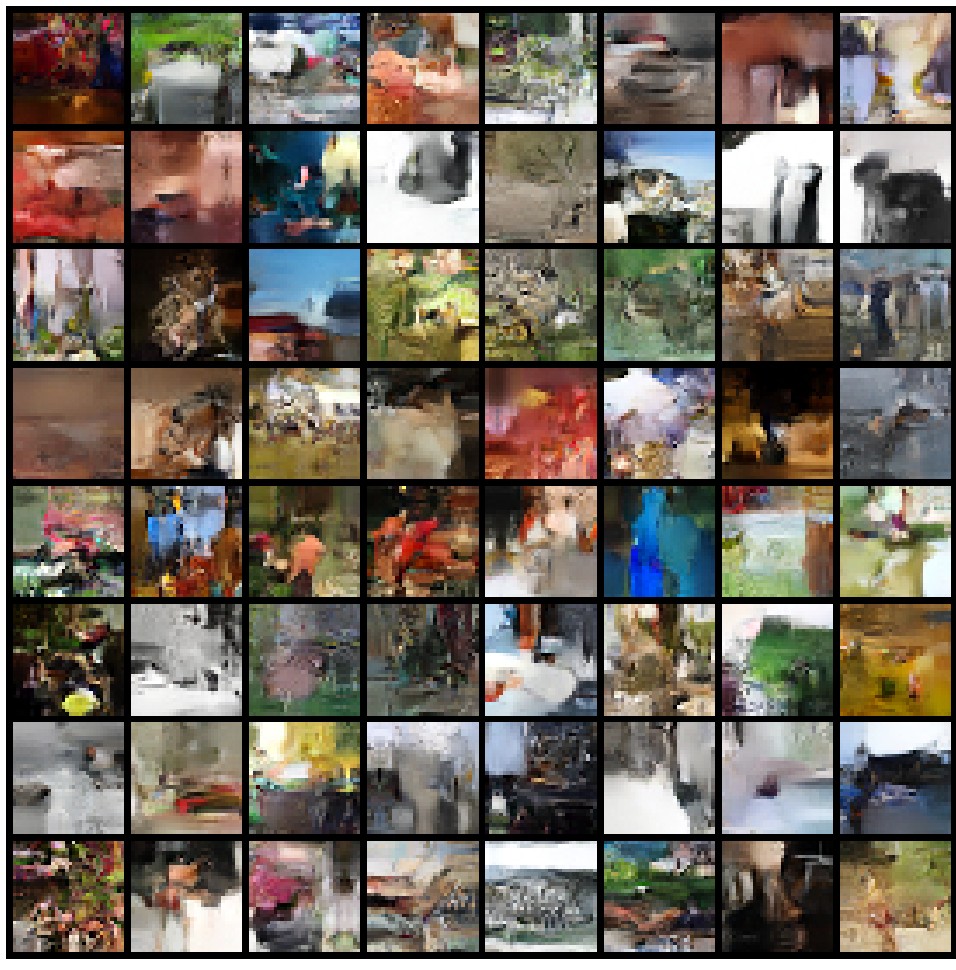

Figure 9: Generation results on the CIFAR-10 dataset.

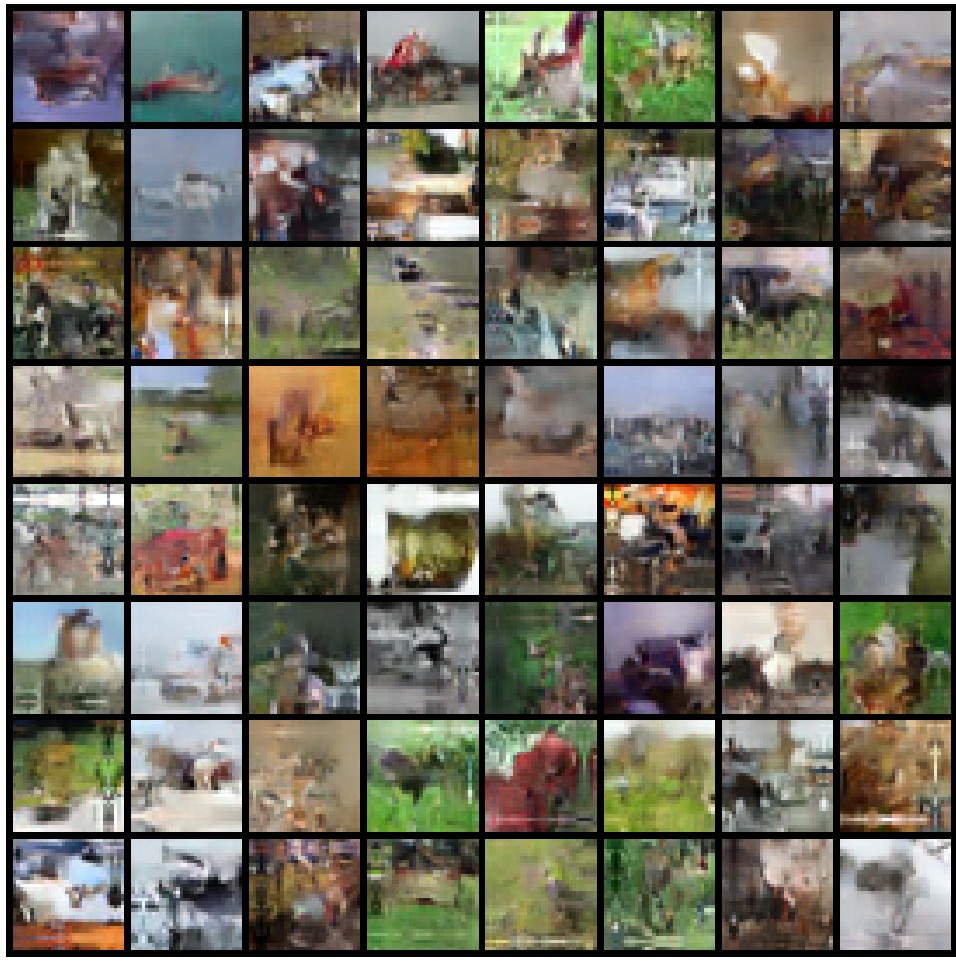

Figure 10: Generation results on the ImageNet dataset.

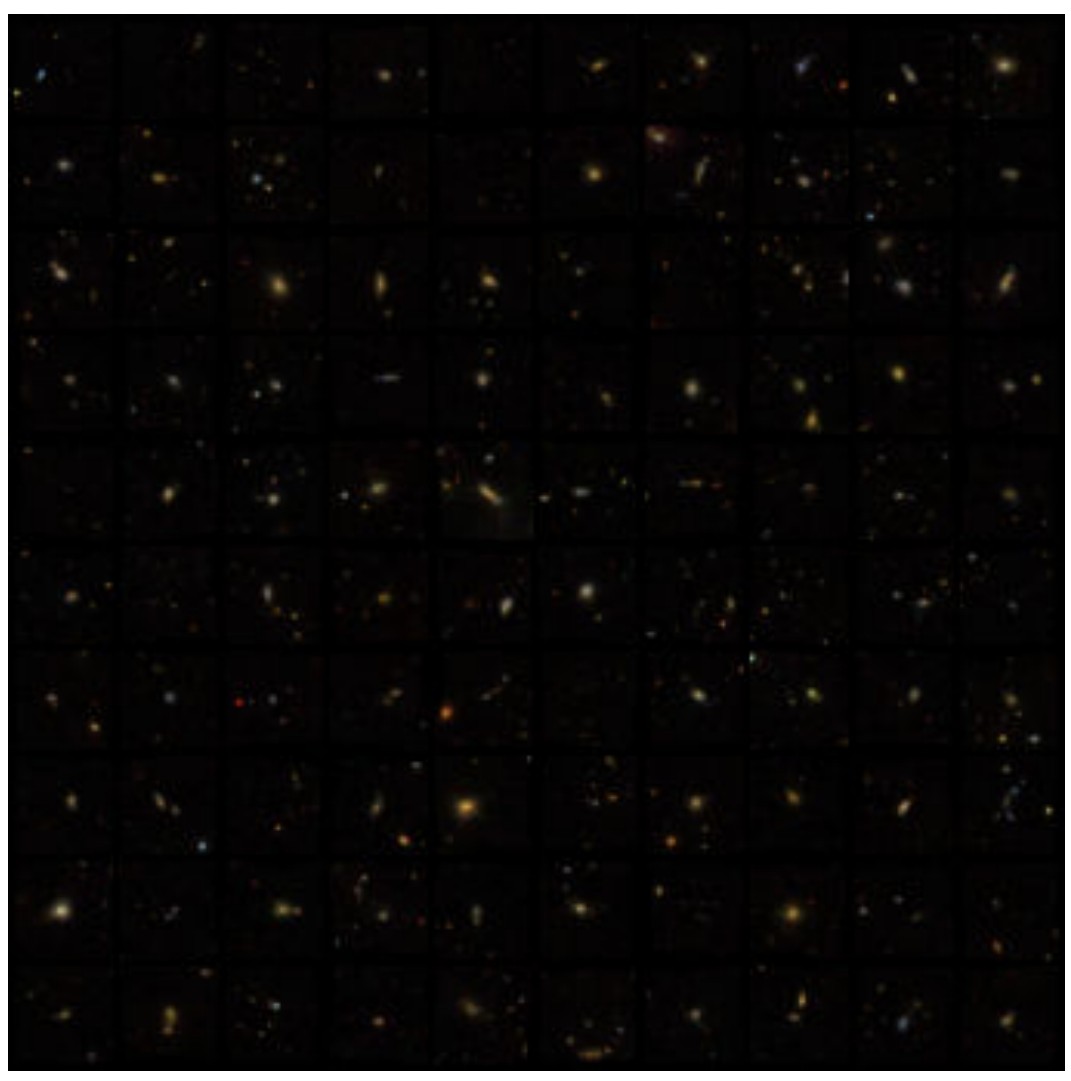

Figure 11: Generation results on the Galaxy dataset.

