# OpenReview forum: "CDFlow: Building Invertible Layers with Circulant and Diagonal Matrices"
_NeurIPS.cc/2025/Conference — NeurIPS 2025 poster_

### Official Review · Reviewer_GMUr · 2025-06-27

**Clarity:** 3
**Significance:** 3
**Originality:** 2
**Rating:** 5
**Confidence:** 4

**Summary:**

This paper presents a new efficient parameterization for linear mixing layers in normalising flow architectures. It relies on a circular-diagonal decomposition of general matrices and leveraging the FFT for efficient matrix inverses and Jacobion computation required in flow models. The resulting speedup is significantly more scalable, where the matrix inversion in the inverse flow direction goes from O(n^3) to O(m n log n) as it is not bottlenecked by the FFT, and the log determinant computation becomes O(mn) due to the simplicity of Jacobians of diagonal and circulant matrices. Empirical results on natural image data with and without periodic features show that it achieve competitive or even superior performance to alternative mixing layer models, while being computationally efficient allowing significant speedup of the training process.

**Questions:**

- It is not clear how to choose the m value in a systematic way, is this just done heuristically? If there are any heuristic/mathematical guidelines please do include this in the method/discussion
- Could you clarify the choices of the baselines hyperparameters, e.g. the Woodbury and butterfly settings described in 4.3?
Are the same baseline hyperparameters for the linear transform layers used in 4.1 and 4.2? This should be stated clearly at the start of 4 if all experiments use similar baseline setups including hyperparameters.
- A note: In the abstract, I realise nowhere the meaning of “m” is mentioned despite being in the complexity expression (related to number of diagonal and circulant matrices in the decomposition). This should be added for clarity and standalone role of the abstract.

Clarification on these points will help me assess whether the framework proposed actually has considered these design and thought points and can help me adjust the score ratings.

**Ethical Concerns:**

["NO or VERY MINOR ethics concerns only"]

**Final Justification:**

The authors' rebuttal addresses my concerns and includes novel results from additional experiments to address the image-only concern in terms of experimental evaluations. Improvements in the explanation in the text are proposed as well, and I increase the overall quality (assuming these are executed on).

**Limitations:**

It would be good to mention avenues for further work and limitations such as the scaling of m required to achieve sufficient performance in general compared to dense transformations, or some discussion of the trade off between performance and the choice of m. I will be able to boost my review scores if a good discussion aspect is added.

**Paper Formatting Concerns:**

N.A.

**Quality:**

4

**Strengths And Weaknesses:**

Strengths:
- Simple but practical idea that should be easily included in many flow implementation packages and pipelines
- Proven idea of circulant-diagonal decomposition used in other fields like variational Gaussian processes, e.g. [Jensen, Kristopher, et al. "Scalable Bayesian GPFA with automatic relevance determination and discrete noise models." Advances in Neural Information Processing Systems 34 (2021): 10613-10626.]
- Competitive performance and evidence of advantages in structured datasets with minimal complexity overhead

Weaknesses:
- Experiments only consider image data
- Hyperparameter choice of the method (number of matrices in the decomposition) seems a bit obscure

---

> ### Author Rebuttal · Authors · 2025-07-31
>
> We sincerely thank the reviewer for the detailed comments and constructive suggestions. We greatly appreciate your recognition of the practical value of our method, its relevance to prior structured decompositions (e.g., circulant-diagonal in Gaussian processes), and its empirical performance on structured datasets. Below, we address each of the raised concerns.
>
> >Proven idea of circulant-diagonal decomposition used in other fields……
>
> We appreciate the reviewer’s insightful remark regarding the broader usage of circulant-diagonal decomposition. As exemplified by [1], this technique has been successfully applied in variational Gaussian processes, further validating its generality and effectiveness. We will add this reference to the Related Work section to acknowledge its relevance and to better position our method within the wider landscape of structured matrix approaches.
>
> >Weakness 1:Experiments only consider image data
>
> To assess broader applicability, we extended our evaluation to the flow matching task by integrating our CDLinear layer into a 5-layer MLP, forming a backbone we term CDMLP. This replaces standard linear layers with our circulant-diagonal decomposition design. Experiments on synthetic 2D datasets (chessboard, moons, concentric circles) show that CDMLP achieves comparable or better performance (MSE, FID, NLL) than MLP, while reducing parameters by ~99×. We will include these results and corresponding generated samples in the Appendix to highlight the generality of our operator beyond image modeling.
>
> Table 1. Flow matching results on synthetic 2D datasets (chessboard, moons, concentric circles). CDMLP uses our proposed CDLinear layers (with $m=2$). Compared to a vanilla MLP baseline, CDMLP achieves competitive or better accuracy with significantly fewer parameters.
>
> | Dataset       | Method     | Params | MSE ↓ | FID ↓   | NLL ↓    |
> |----|----|----|---|-----|------|
> | Chessboard   | CDMLP   | 8,453  | 3.810  | 0.0115   | 7.0425    |
> | Chessboard    | MLP        | 791,042| 3.796  | 0.0062   | 10.6853   |
> | Moons        | CDMLP   | 8,453  | 1.842  | 0.0016   | 31.0329   |
> | Moons       | MLP        | 791,042| 1.854  | 0.0022   | 24.1885   |
> | Circles      | CDMLP   | 8,453  | 2.482  | 0.0069   | 15.4922   |
> |  Circles      | MLP       | 791,042| 2.433  | 0.0019   | 42.4687   |
>
> >Weakness 2& Question 1: Hyperparameter choice of the method seems a bit obscure
>
> In our method, each linear transform is decomposed into $2m - 1$ matrices (i.e., $m$ diagonal and $m - 1$ circulant). In our main experiments, we set $m = 2$, which results in 3 structured matrices. This setting reflects a trade-off between model expressivity and efficiency. A larger $m$ increases the approximation power but also incurs higher parameter and computational costs.
> Our objective is to achieve an optimal trade-off among model expressiveness, parameter and computational efficiency, and runtime performance. To this end, we systematically evaluated the influence of varying $m$ on model behavior.
>
> As summarized in Table 2, increasing $m$ on CIFAR-10 yields only marginal improvements in expressiveness, as indicated by BPD values, while incurring increases in both parameter count and computational cost. Table 3 and Table 4 present the runtime measurements for both forward and reverse operations. Although a larger $m$ introduces additional latency, our method remains more efficient than alternative approaches reported in the original paper.
>
> Table 2. Performance and parameter statistics of CDFlow under different values of $m$ on CIFAR-10.
> | Model | Setting | BPD (CIFAR-10)  | DCD Params |
> |-----|------|--------|------|
> | CDFlow | m=2 | 3.31  | 6.9 K      |
> | CDFlow| m=3 | 3.30   | 11.1 K     |
> | CDFlow | m=4  | 3.30   | 15.3 K     |
>
> Table 3. Forward computation time (in milliseconds) for CDLinear layers with different values of $m$, compared against traditional $1 \times 1$ convolutions.
>
> | Channels | $m=2$ | $m=3$ | $m=4$ | $m=5$ | $m=6$ |1x1 Conv|
> |----------|-------|-------|-------|-------|--------|-------|
> | 16     | 0.226 | 0.226 | 0.326 | 0.374 | 0.424  | 0.240    |
> | 32     | 0.323 | 0.405 | 0.547 | 0.690 | 0.837  | 0.432    |
> | 48     | 0.379 | 0.575 | 0.792 | 1.008 | 1.225  | 0.477    |
> | 64     | 0.454 | 0.729 | 1.007 | 1.287 | 1.571  | 0.535    |
> | 80     | 0.557 | 0.909 | 1.267 | 1.625 | 1.983  | 0.679    |
> | 96     | 0.719 | 1.194 | 1.676 | 2.169 | 2.644  | 0.758    |
> | 128    | 0.875 | 1.434 | 1.963 | 2.529 | 3.144  | 0.880    |
>
> Table 4. Inverse computation time (in milliseconds) for CDLinear layers with different $m$ values, compared against $1 \times 1$ convolutions.
>
> | Channels | $m=2$ | $m=3$ | $m=4$ | $m=5$ | $m=6$ |1x1 Conv |
> |----------|-------|------|-------|-------|--------|-------|
> | 16       | 0.170 | 0.229 | 0.291 | 0.353 | 0.416  | 0.195 |
> | 32       | 0.268 | 0.407 | 0.549 | 0.697 | 0.836  | 0.328 |
> | 48       | 0.348 | 0.566 | 0.782 | 0.993 | 1.234  | 0.428 |
> | 64       | 0.428 | 0.706 | 1.011 | 1.264 | 1.544  | 0.511 |
> | 80       | 0.532 | 0.890 | 1.275 | 1.598 | 1.972  | 0.714 |
> | 96       | 0.693 | 1.168 | 1.653 | 2.135 | 2.633  | 0.757 |
> | 128      | 0.849 | 1.397 | 1.940 | 2.492 | 3.054  | 0.935 |
>
> In addition, we conducted a separate evaluation of the log-determinant computation time across varying channel sizes from 16 to 1024. The proposed method demonstrates consistent runtime performance across different values of $m$, with execution time ranging between 0.095 ms and 0.097 ms.
>
> Based on these observations, we adopt $m = 2$ as the default setting for all experiments, including both the original and supplementary ones, unless otherwise specified.
>
> >Question2: Could you clarify the choices of the baselines hyperparameters
>
> To ensure fairness, we adopt consistent backbone settings for the subset of flow models that share a similar multi-scale architecture—namely, Glow, Woodbury, Emerging, and ButterflyFlow—each using 3 blocks and 32 flow steps.. For Woodbury, we follow the original configuration for CIFAR-10 with $d_s = \{16, 16, 8\}$ and $d_c = \{8, 8, 16\}$; for ButterflyFlow, we set $butterfly level=1$ as per its default. Thank you for pointing this out — we will clarify these settings at the beginning of Section 4.
>
> >Question 3: In the abstract, I realise nowhere the meaning of “m” is mentioned……
>
> To improve clarity and ensure that the abstract is self-contained, we will revise the sentence as follows:
> "Furthermore, leveraging the Fast Fourier Transform (FFT), our method reduces the time complexity of matrix inversion from $O(n^3)$ to $O(mn \log n)$ and matrix log-determinant from $O(n^3)$ to $O(mn)$, where $n$ is the input dimension and $m$ denotes the number of diagonal matrices used in the decomposition, which are combined with $m-1$ circulant matrices to construct each CDLinear layer."
>
> >Limits：It would be good to mention avenues for further work and limitations……
>
> As discussed earlier in our response regarding the choice of $m$, we now aim to further explore the representational capacity of this decomposition scheme. To this end, we reproduced the experiments from Appendix A.5 of [2] and included results using circulant matrices and diagonal-circulant-diagonal ($\text{DCD}$) matrices. As shown in Table 5, the circulant matrix achieves faster convergence and yields performance closest to the full matrix. Moreover, the performance of the $\text{DCD}$ method significantly surpasses that of $\text{LU}$ and the individual $\text{L}$ and $\text{U}$ decompositions, highlighting the expressive potential of this class of matrix decompositions.
>
> Table 5. Mean NLL ± 95% CI on synthetic data (step 5000 / 10000)
> | Type  | 5000 NLL     | 10000 NLL    |
> | ----- | ------------ | ------------ |
> | full  | 49.11 ± 0.01 | 48.32 ± 0.12 |
> | tril  | 51.91 ± 0.34 | 50.89 ± 0.18 |
> | triu  | 50.27 ± 0.38 | 49.61 ± 0.73 |
> | trilu | 51.85 ± 0.49 | 51.04 ± 0.22 |
> | circ  | 49.07 ± 0.24 | 48.49 ± 0.39 |
> | dcd   | 50.18 ± 0.38 | 49.21 ± 0.12 |
>
> From a theoretical perspective, our decomposition can approximate any linear operator, and since convolutions can be rewritten as matrix multiplications, our method is in principle capable of representing any convolutional transformation. However, in the context of normalizing flows, this capacity is constrained by the need to compute the matrix inverse and log-determinant during inference. When convolution is converted to a matrix multiplication, the resulting weight matrix is typically non-square, making the determinant and inverse undefined in the classical sense.
>
> Addressing this limitation is a key direction for future work. One possible avenue is to explore extensions of determinant and inverse computation to non-square matrices, such as using the the Moore–Penrose pseudoinverse [3] orRadic determinant[4]. By incorporating these tools, we hope to generalize our method to a broader class of transformations, including non-square and structured convolutional layers, thereby expanding the applicability of our framework to arbitrary function classes.
>
> Thank you again for your insightful feedback. We hope these clarifications and additions address your concerns and provide a clearer view of the design and potential of our framework. We would greatly appreciate your reconsideration of the manuscript in light of these improvements.
>
> [1]Jensen, Kristopher, et al. "Scalable Bayesian GPFA with automatic relevance determination and discrete noise models." NeurIPS'21: 10613-10626.
>
> [2]Chao, Chen-Hao, et al. "Training energy-based normalizing flow with score-matching objectives." NeurIPS'23: 43826-43851.
>
> [3]Penrose, Roger. "A generalized inverse for matrices." Mathematical proceedings of the Cambridge philosophical society. Vol. 51. No. 3. Cambridge University Press, 1955.
>
> [4]Radic, M. "A definition of the determinant of a rectangular matrix." Glasnik Matematicki 1.21 (1966): 17-22.

---

> > ### Comment · Reviewer_GMUr · 2025-08-01
> > **Rebuttal response comment**
> >
> > Thank you for the detailed response to my concerns. The application to synthetic 2D datasets is a great addition outside of the (higher dimensional) image data and a great add. The hyperparameter sweep results look good and justify the hyperparameter choice in the paper, as well as the proposed text changes to more explicitly mention experiment details look good.

---

### Official Review · Reviewer_fQVx · 2025-07-01

**Clarity:** 4
**Significance:** 3
**Originality:** 3
**Rating:** 5
**Confidence:** 4

**Summary:**

The authors present a new flow architecture that uses a product of diagonal and circulant matrices for the weight matrices. This enables parsimonious representation of weight matrices, and efficient forward, backward, and determinant calculation. The authors also show that this computation allows for good density estimation, and is often computationally more efficient than existing competing architectures.

**Questions:**

Questions/Comments

 - In the abstract, what is meant by "This decomposition provides a compact representation?" I don't understand what the phrase "compact representation" means here. Could you please explain?
 - In line 59, "...storing a collection of circulant and diagonal vectors." The vectors are not circulant or diagonal, the matrices are, correct?
 - In line 63, "By utilizing the Fast Fourier Transform (FFT), the determinant computation is simplified to $O(n)$." This sentence is not right. The runtime of the determinant computation doesn't follow from the FFT. It follows because you store the eigenvalues of the circulant layers directly.
 - Related to the point above, line 63 "the determinant of a diagonal matrix equals the product of its diagonal elements, and the determinant of a circulant matrix is the product of its eigenvalues." This sentence is very odd. The determinant of every diagonalizable matrix is the product of its eigenvalues. It is not a special property of circulant matrices, and is the reason a diagonal matrix's determinant is the product of its diagonal entries.
 - In the paragraph in lines 189 - 194, I don't understand what is being said. What is "the invertible convolutional layer?" Are you referring to the circulant layers? Are you building a specific convolution kernel via the product of diagonal and circulant matrices? How is this similar to the 1x1 convolutions from Dinh et al.? I think this point needs to be clarified considerably, especially given the discussion in section 4.3

Typos:
 - In line 134, there is a missing space between the period and reference.
 - In line 151, the t in The should not be capitalized.
 - In line 166 "...its inverse is to compute the reciprocal..." There is a typo here.
 - In line 169 "is similar to diagonal matrix" should be "is similar to a diagonal matrix."
 - In line 171, the word "with" is unnecessary.
 - Line 203, "handle complex and complex distributions." I don't understand what is meant here.
 - Line 252, there is a space missing after the word model.

**Ethical Concerns:**

["NO or VERY MINOR ethics concerns only"]

**Final Justification:**

The paper is quite nice. My criticisms of the paper were quite minor, and the authors have addressed them all. My recommendation stands.

**Limitations:**

Yes.

**Quality:**

3

**Strengths And Weaknesses:**

Strengths:
 - The paper is well-written and easy to follow.
 - The core idea is simple, elegant and well-executed.
 - The presentation of the mathematics is clear and intuitive.
 - The numerical experiments, especially those presented in Section 4.3, clearly valorizes the work.

Weaknesses:
 - The work has many small typos scattered throughout. Please see the questions section.
 - The experiments show that CDFlow performs well in practice, but the gains in terms of density estimation are modest, at best.

---

> ### Author Rebuttal · Authors · 2025-07-31
>
> We sincerely thank the reviewer for the thoughtful and constructive comments. We greatly appreciate the positive feedback on the clarity of our writing, the elegance of our method, and the quality of the mathematical presentation and experiments. Below, we provide point-by-point responses to each of the raised concerns. All suggested modifications have been implemented in the revised manuscript.
>
> >Weaknesses 2: CDFlow performs well in practice, but the gains in terms of density estimation are modest.
>
> We acknowledge the reviewer’s observation. Indeed, our primary focus is not noly on boosting bits-per-dimension (BPD) or perceptual quality, but also on improving the parameter efficiency and computational complexity of flow-based models. Specifically, our method reduces the parameter count of a dense linear layer from $O(n^2)$ to $O(mn)$, where $m$ is a small constant (set to 2 in our implementation).  Moreover, by leveraging the structure of diagonal and circulant matrices, we reduce the computational complexity of matrix inversion from $O(n^2)$ to $O(n \log n)$. These gains are especially impactful in large-scale or resource-constrained deployments, and the efficiency has been demonstrated in Section 4.3.
>
> >Question 1: "The term ‘compact representation’ in the abstract is vague."
>
> We agree and have revised the abstract to improve clarity. The updated phrasing reads:
>  “This decomposition provides a parameter- and computation-efficient formulation, reducing the parameter complexity from $\mathcal{O}(n^2)$ to $\mathcal{O}(mn)$, while preserving the transformation ability of full linear layers.”
>  This avoids ambiguous terms and emphasizes the structural efficiency without overstating representational properties.
>
> >Question 2: In line 59, "...storing a collection of circulant and diagonal vectors."
>
> For the wording in line 59, we understand the reviewer’s concern. While the matrices involved are circulant or diagonal, our implementation stores only the generating vectors for each: the diagonal elements of the diagonal matrix and the eigenvalue vector of the circulant matrix (i.e., the DFT of its defining vector). Therefore, we retain the term “vectors” intentionally and have clarified this point in the manuscript to avoid confusion.
>
> >Question 3:  in line 63, “By utilizing the Fast Fourier Transform (FFT), the determinant computation is simplified...,”
>
> With respect to the sentence in line 63, we acknowledge the imprecision in our original phrasing. The actual runtime gain does not come directly from the FFT itself during inference, but from the fact that the eigenvalues of circulant matrices are computed once during initialization using an $O(n \log n)$ FFT, and are subsequently reused. We have modified the sentence to:
> "By performing an $O(n \log n)$ FFT during initialization, we precompute the eigenvalues of circulant matrices, allowing the log-determinant to be computed at runtime in $O(n)$ time."
>
> >Question 4:  The determinant of a diagonal matrix equals the product of its diagonal elements...’ is misleading.
>
> In a related clarification, the sentence, “the determinant of a diagonal matrix equals the product of its diagonal elements, and the determinant of a circulant matrix is the product of its eigenvalues,” was indeed misleading. As correctly noted, the determinant of any diagonalizable matrix is the product of its eigenvalues. We have revised this to:
> "The determinant of a diagonalizable matrix equals the product of its eigenvalues. In our case, the stored diagonal and circulant vectors correspond to these eigenvalues, allowing for efficient log-determinant computation."
>
> >Question 5:  Lines 189–194 and the term ‘invertible convolutional layer’ are vague.
>
> The term was used imprecisely and has now been explicitly tied to the “CD-Convolution layer” illustrated in Figure 1. We revised the sentence to:
> "The invertible convolutional layer, referred to as the CD-Convolution layer in Figure 1, leverages special weight matrices—specifically circulant and diagonal matrices—to enable efficient computation."
>
> Further, in response to the comparison with 1×1 convolutions, our method simulates such convolutions by matrix multiplication, where the weight matrix is approximated using the diagonal-circulant decomposition. While 1×1 convolutions (e.g., in Kingma & Dhariwal, 2018) use dense $n \times n$ kernels requiring $O(n^2)$ parameters, our approach only requires $O(mn)$ parameters. Moreover, log-determinant and inverse operations are significantly more efficient in our design, with $O(n)$ and $O(n \log n)$ complexities respectively, compared to $O(n^3)$ in standard 1×1 convolutions and $O(n^2)$ in LU-decomposed ones. These theoretical benefits are reflected in the runtime advantages shown in Section 4.3.
>
> >Typos
>
> Lastly, we sincerely thank the reviewer for identifying several typos. We have carefully reviewed and corrected the following issues in the revised manuscript:
> - Line 134: Added missing space.
> - Line 151: Fixed capitalization of “The.”
> - Line 166: Corrected grammatical structure.
> - Line 169: Inserted missing article.
> - Line 171: Removed unnecessary “with.”
> - Line 203: Rewrote unclear phrase.
> - Line 252: Inserted missing space.
>
> We believe these revisions improve both the clarity and technical accuracy of the paper. Thank you again for your valuable feedback and for acknowledging the overall technical solidity and impact of our work. We hope the revised version addresses all concerns satisfactorily.

---

> > ### Comment · Reviewer_fQVx · 2025-08-05
> >
> > These changes all sound good to me. Thank you for the response. I shall keep my score.

---

### Official Review · Reviewer_bzQ3 · 2025-07-02

**Clarity:** 3
**Significance:** 2
**Originality:** 1
**Rating:** 4
**Confidence:** 4

**Summary:**

The paper presents and explores a flow architecture, the distinctive feature of which is the usage of the product of interleaving circulant and diagonal matrices as the weights for 1x1 convolutional layers, constituting the non-linear transformation of the coupling layer. The main advantages of this approach are the reduced time complexity for logdet and inverse computation, as well as decreased space consumption.

**Questions:**

Please see **action items** and **possible action items** in “Weaknesses”. I will be ready to accept the paper if the first three weaknesses are thoroughly addressed. Moreover, I have some smaller questions:

1. Could you please provide experimental results (both regarding sampling quality and runtime efficiency) for different choices of hyperparameter m?
2. What can be done to represent the non-square weight matrix?
3. Could you please explain why there is such a large margin between the proposed method and other methods, especially in logdet computation? Some of the methods have the same theoretical complexity, why are they so much slower?
4. Did you try using the initialization scheme and the placement of non-linearities proposed in [1]? I think it might increase the quality of the final model.

**Ethical Concerns:**

["NO or VERY MINOR ethics concerns only"]

**Final Justification:**

During the rebuttal phase, the authors have resolved W1 and W2 from my initial comment and have promised to report mean and std of all their experiments in the final version of the paper, which was the main point of W3. W4 was also addressed in their comment from the theoretical point of view, although the experimental evaluation (with reproduction of the results of the discussed paper) would add value to their paper (and the code for reproduction would be useful for the community). The authors have also answered my questions, although I have now noticed that I missed the opportunity to ask specifically for the times for circulant layers without diagonal ones (i.e. for m = 1). Overall, I feel that conceptually this work is incremental and lacks originality, as it just mixes together two already well-known concepts. However, the experimental results are solid and convincing, and the authors actively participated in the rebuttal. Thus, I am raising "quality" score to 3 and the overall score to 4.

**Limitations:**

The authors adequately addressed the limitations and potential negative societal impact of their work.

**Quality:**

3

**Strengths And Weaknesses:**

**Strengths**:
1. The paper is written very clearly, with all the necessary prerequisites explained in Section 2 (“Background”).
2. In order to prove the superiority of their method, authors do not only compute the theoretical complexity, but provide experimental time measurements in Section 4.3 as well.
3. The proposed method, indeed, has a competitive time complexity, both in theory and in practice.

**Weaknesses**:

Here are some conceptual weaknesses of the paper:

1. As the diagonal-circulant neural networks have already been proposed in [1], the novelty of the paper lies exclusively in applying such networks to the construction of a flow model. While the efficiency of this approach is confirmed experimentally, the trade-off between the number of parameters of the flow layers and the quality of the resulting network is not explored. According to [2], this might be a non-negligible trade-off. **Possible action items**: 1) Please check if it would be helpful to rearrange the results from Table 2 into a graph (or two graphs) with number of parameters (or runtime measurements from Figure 3) on one axis and quality on the other. 2) Could you reproduce the experimental setting of Appendix A.5 in [2] with the proposed method?
2. The experimental section of the paper is scarce. While it is clearly shown that the computation of the Jacobian logdet is much faster than its alternatives, especially for large numbers of channels, which might indeed speed the training up, the sampling quality from Table 2 is average. **Possible action item**: The paper would have benefited much from applying the proposed layers to a real experimental setting, e.g. in a flow matching model or comparing to Woodbury in a setup similar to [3]. Speed up and quality measurements obtained this way would be much more valuable and would help other researchers to choose the appropriate method of flow layer compression.
3. A couple of words about the presented results: while I understand that the proposed method does not have to be SOTA, as its main goal is efficiency, the results from Table 2 and Figure 2 are not very convincing. In Table 2 the results are presented without confidence intervals. Also, it is unclear why MAF and i-ResNet are not evaluated on ImageNet and even more methods are not evaluated on Galaxy. In Figure 2 the images are too small to see any difference, but to me it seems that images from the proposed method have some strange artifacts which other methods do not have. **Action items**: Please report the results for all methods, add confidence intervals and show larger pictures.
4. The authors did not mention some very relevant related articles, which is alarming, as their work could have benefited much from using tips for training from [1] (this paper answers the questions of expressivity and efficient training methods of networks with circulant-diagonal layers) and comparing their method to a close in time complexity method from [4] (with both Jacobian determinant and inverse computation complexity being O(nlogn)). Also, similar parameterizations of linear layers as the product of circulant and diagonal matrices were proposed in [5], [6]. **Action item**: Please compare to [4] (please correct me if they are not comparable).

Some small inaccuracies in the manuscript:

1. In equation (3) there are L - 1 terms, while there should be L, with the first one being df_1 / dx.
2. Lines 125-127: it is written that the proposed complexity O(n log n) is more efficient than O(dn) from Woodbury, where d is a constant, which is not true, especially considering the fact that the proposed complexity is actually O(mn log n), where m is a constant.
3. In Figure 2, there is a typo in the name of “Woodbury”.

[1] Araujo, Alexandre, et al. "Understanding and training deep diagonal circulant neural networks." ECAI 2020. IOS Press, 2020. 945-952.

[2] Chao, Chen-Hao, et al. "Training energy-based normalizing flow with score-matching objectives." Advances in Neural Information Processing Systems 36 (2023): 43826-43851.

[3] Lee, Jaewoo, et al. "Differentially private normalizing flows for synthetic tabular data generation." Proceedings of the AAAI Conference on Artificial Intelligence. Vol. 36. No. 7. 2022.

[4] Karami, Mahdi, et al. "Invertible convolutional flow." Advances in Neural Information Processing Systems 32 (2019).

[5] Cheng, Yu, et al. "An exploration of parameter redundancy in deep networks with circulant projections." Proceedings of the IEEE international conference on computer vision. 2015.

[6] Moczulski, Marcin, et al. "Acdc: A structured efficient linear layer." arXiv preprint arXiv:1511.05946 (2015).

---

> ### Author Rebuttal · Authors · 2025-07-31
>
> We sincerely thank the reviewer for the constructive suggestions and valuable references. Below, we provide detailed responses to each of the raised concerns and questions.
>
> >Weakness 1: Parameter counts and quality
>
>  We revised Table 1 to include model parameter counts and added supplementary experiments.
>
> Table 1. Performance comparison of various flow-based models. BPD (bits per dimension) is reported along with the number of model parameters.
> |Model|Params|CIFAR-10 BPD|Params|ImageNet BPD|Params|Galaxy BPD|
> |--|--|--|--|--|--|--|
> |Ours|44.2M|3.31|44.2M|4.04|6.25M|1.92|
> |Butterfly|44.4M|3.33|44.4M|4.09|6.39M|1.95|
> |Emerging|46.6M|3.34|44.0M|4.09|6.68M|1.98|
> |Woodbury|45.3M|3.35|45.3M|4.09|6.68M|2.01|
> |Glow|44.2M|3.35|66.2M|4.09|6.25M|2.02|
> |MAF|105.0M|4.31|—|—|—|—|
> |RealNVP|6.4M|3.49|46.2M|4.28|6.38M|2.11|
> |Residual Flows|25.2M|3.28|47.1M|4.01|5.58M|3.60|
> |i-DenseNet|24.9M|3.25|47.0M|3.98|6.07M|4.06|
> |i-ResNet|25.0M|3.45|29.7M|4.05|6.88M|2.04|
>
> We reproduced the experiments from A.5 of [1], adding results with circulant and diagonal-circulant-diagonal (DCD) matrices.(Table 2) Circulant matrices converge faster and closely match full matrix performance and DCD variant outperforms LU and its individual L and U components, demonstrating strong expressive power.
>
> Table 2. Mean NLL ± 95% CI on synthetic data (step 5k / 10k)
> |Type|5k NLL|10k NLL|
> |-|-|-|
> |full|49.11±0.01|48.32±0.12|
> |tril|51.91±0.34|50.89±0.18|
> |triu|50.27±0.38|49.61±0.73|
> |trilu|51.85±0.49|51.04±0.22|
> |circ|49.07±0.24|48.49±0.39|
> |dcd|50.18±0.38|49.21±0.12|
>
> >Weakness 2:The experimental section of the paper is scarce.
>
> To demonstrate broader applicability, we replaced standard linear layers with CDLinear in a 5-layer MLP for flow matching on 2D toy datasets(Table 3). Our method achieves comparable or better MSE, NLL, and FID with significantly fewer parameters.
>
> Table 3. Flow Matching Model Results on 2D-toy Datasets
> |Dataset|Model|Params|MSE↓|FID↓|NLL↓|
> |---|---|---|---|---|---|
> |Chessboard|CDMLP|8.5K|3.81|0.0115|7.0425|
> |Chessboard|MLP|791K|3.796|0.0062|10.6853|
> |Moons|CDMLP|8.5K|1.842|0.0016|31.0329|
> |Moons|MLP|791K|1.854|0.0022|24.1885|
> |Circles|CDMLP|8.5K|2.482|0.0069|15.4922|
> |Circles|MLP|791K|2.433|0.0019|42.4687|
>
> In addition, [2] did not provide open-source code, which made it difficult to perform a direct comparison between the Woodbury approach and our method under a consistent experimental setup within the limited time available.
>
> >Weakness 3: A couple of words about the presented results.
>
> We did not report classical confidence intervals, as none of the baseline papers did, reflecting the limited use of statistical testing in deep learning research. Unlike traditional statistical experiments, deep learning models are typically trained multiple times on the same fixed dataset, and the resulting outcomes do not satisfy the assumption of i.i.d. samples. Consequently, conventional tests such as the $t$-test or ANOVA are not applicable. More rigorous comparisons require resampling-based methods as discussed in [3]. However, these methods are often impractical in deep learning scenarios, as they demand a large number of repeated experiments and careful control of confounding variables for each model, making them difficult to incorporate into standard evaluation pipelines.
> As a result, the common practice is to report the mean and standard deviation across multiple runs to assess model stability. For example, on CIFAR-10, we conducted three trials and achieved a BPD of $3.31 \pm 0.004$, demonstrating the robustness of our approach.
>
> >Weakness 4: The authors did not mention some very relevant related articles.
>
> Thank you for pointing out the missing references. We will add the suggested citations and related discussions.
>
> [4] focuses on symmetric and circulant convolutions for coupling layers, while our method approximates arbitrary convolutions and fully connected layers by decomposing weight matrices into interleaved circulant and diagonal components, enabling fast matrix multiplication.
>
> Besides, [4] github repo is empty, making reproduction and fair comparison infeasible. We therefore focus on theoretical distinctions.
>
> In terms of complexity, [4] relies on 2D FFTs over spatial dimensions ($\mathcal{O}(chw \log(hw))$), while ours uses 1D FFTs over channels ($\mathcal{O}(chw \log c)$ for convolution, $\mathcal{O}(c \log c)$ for log-determinants). As stated in Lines 83–86, the Jacobian of a linear layer equals the weight matrix $W$, so $\log|\det(W)|$ scales with $c$, not $h$ or $w$.
>
> In summary, [4] tackles a different problem and is not directly comparable to our approach.
>
> >Some small inaccuracies in the manuscript.
>
> Thank you for pointing out the issues. We have revised Eq.3 as $\log |\partial f_\theta / \partial x| = \log |\partial f_1 / \partial x| + \sum_{i=2}^L \log |\partial f_i / \partial f_{i-1}|$. Additionally, we have clarified the description in Lines 125–127: Although Woodbury's theory has better complexity than ours, in actual operation, they need to transform the input multiple times, so the actual efficiency is not as good as ours. Lastly, we have corrected the typo in the name "Woodbury" in Figure 2.
>
> >Q1 : provide experimental results for different choices of hyperparameter m.
>
> Our goal is to balance expressiveness, efficiency, and runtime. We evaluated the impact of varying $m$ on CIFAR-10(Table 4), larger $m$ brings only minor BPD improvements but increases parameter count and computational cost. Runtime results(Tables 5 and 6) confirm that our method remains more efficient than prior alternatives, even as $m$ grows. We further evaluated log-determinant computation across channel sizes from 16 to 1024. The average runtime remains stable (0.095–0.097 ms) across different $m$ values, confirming the scalability of our approach. Therefore, we set $m = 2$ as the default for all experiments unless noted otherwise.
>
> Table 4. Performance and parameter statistics of CDFlow under different values of $m$ on CIFAR-10.
> |Model|m|BPD(CIFAR-10)|DCD Params|
> |--|--|--|--|
> |CDFlow|2|3.31|6.9K|
> |CDFlow|3|3.30|11.1K|
> |CDFlow|4|3.30|15.3K|
>
> Table 5. Forward computation time (ms) for CDLinear layers with different values of $m$, compared against $1 \times 1$ convolutions.
>
> |Channels|$m=2$|$m=3$|$m=4$|$m=5$|$m=6$|1x1 Conv|
> |--|--|--|--|--|--|--|
> |16|0.226|0.226|0.326|0.374|0.424|0.240|
> |32|0.323|0.405|0.547|0.690|0.837|0.432|
> |48|0.379|0.575|0.792|1.008|1.225|0.477|
> |64|0.454|0.729|1.007|1.287|1.571|0.535|
> |80|0.557|0.909|1.267|1.625|1.983|0.679|
> |96|0.719|1.194|1.676|2.169|2.644|0.758|
> |128|0.875|1.434|1.963|2.529|3.144|0.880|
>
>
> Table 6. Inverse computation time (ms) for CDLinear layers with different $m$ values, compared against $1 \times 1$ convolutions.
>
> |Channels|$m=2$|$m=3$|$m=4$|$m=5$|$m=6$|1x1 Conv|
> |-|-|-|-|-|-|-|
> |16|0.170|0.229|0.291|0.353|0.416|0.195|
> |32|0.268|0.407|0.549|0.697|0.836|0.328|
> |48|0.348|0.566|0.782|0.993|1.234|0.428|
> |64|0.428|0.706|1.011|1.264|1.544|0.511|
> |80|0.532|0.890|1.275|1.598|1.972|0.714|
> |96|0.693|1.168|1.653|2.135|2.633|0.757|
> |128|0.849|1.397|1.940|2.492|3.054|0.935|
>
> >Q2 : non-square weight matrix.
>
> To handle non-square weights, we apply a block-wise partitioning strategy [5] to the first circulant matrix only. Specifically, a matrix $\mathbf{C} \in \mathbb{R}^{d_1 \times d_2}$ is divided into $q_1 = \lceil d_1/p \rceil$ by $q_2 = \lceil d_2/p \rceil$ blocks of size $p \times p$, each treated as a circulant matrix. Dimensions not divisible by $p$ are automatically padded by replication. This converts the original multiplication into multiple independent square circulant multiplications, enabling efficient FFT-based computation. The remaining matrices (e.g., diagonal and secondary circulant) remain square for simplicity and efficiency.
>
> >Q3 : a large margin especially in logdet computation.
>
> Upon rechecking our experiments, we found that the previously reported runtime for Emerging was affected by ambiguities in its source code, which caused our measurement to include an extra 1×1 convolution. Similarly, the Butterfly implementation bundled forward propagation time. After careful retesting, we observed that Emerging’s actual average log-determinant runtime is stable at 0.123 ms, while our method achieves 0.107 ms. The slower runtime of Emerging stems from its need to compute two autoregressive convolutional kernels. For Butterfly, the increased latency arises from its decomposition depth scaling with the input dimension; for fairness, we followed the original setting. All other results remain correct. Notably, both WTrans and MEWTrans involve multiple input transformations, whereas our method provides consistently efficient log-determinant computation.
>
> >Q4 : initialization scheme and the placement of non-linearities.
>
> We did not stack multiple linear layers and therefore did not apply activation functions. However, we agree that stacking linear layers is a promising approach to enhance model expressiveness. Regarding the initialization strategy, we observed no significant performance difference on CIFAR-10 in our experiments.
>
> Thank you again for your valuable feedback and for acknowledging the overall technical solidity and impact of our work. We hope that the revisions and clarifications provided here effectively address your concerns and contribute to a more favorable evaluation.
>
> [1] Chao, Chen-Hao, et al. "Training energy-based normalizing flow with score-matching objectives." NeurIPS'23.
>
> [2]  Lee, Jaewoo, et al. "Differentially private normalizing flows for synthetic tabular data generation." AAAI'22.
>
> [3] T. G. Dietterich, "Approximate Statistical Tests for Comparing Supervised Classification Learning Algorithms," in Neural Computation,1998
>
> [4] Karami, Mahdi, et al. "Invertible convolutional flow." NeurIPS'19.
>
> [5]Ding, Caiwen, et al. "Circnn: accelerating and compressing deep neural networks using block-circulant weight matrices." MICRO’17

---

> > ### Comment · Reviewer_bzQ3 · 2025-08-03
> >
> > Thank you for conducting additional experiments!
> >
> > W1, parameter counts: the updated table is helpful: now, it is easy to see that CDFlow, Emerging, Butterfly and Woodbury are clustered together both by the size of the model and by quality, with only i-ResNet and RealNVP achieving similar quality on all benchmarks from the class of models with smaller sizes.
> >
> > W1, NLL on synthetic task: do I understand correctly that adding diagonal layers to a circulant layer does not improve the expressivity, as measured by this benchmark? Does it mean that using only one circulant matrix, without diagonal ones, instead of m = 2 in other experiments might also be beneficial? Have you conducted such experiments? Were they already conducted in literature?
> >
> > W2: Thank you for considering another dataset. Unfortunately, the new experiment raises more questions than it answers, i.e. 1) What is the quality of the fully connected network on other reported datasets? 2) What is the quality of other methods on this dataset? Overall, my comment was about the lack of in-depth analysis of differences between the proposed layer parameterization and existing ones. The proposed method should be tested in such an experimental setup, so that the results would help researchers and practitioners clearly distinguish between different flow layer parameterizations.
> >
> > W3: I agree that reporting classical confidence intervals might be difficult. In fact, by this comment I wanted to ask you to report exactly the mean and std of several experiments with different random seeds – as you mentioned, it is common practice. However, here you reported mean and std only for one experiment. Sorry for misleading you with my initial comment! Could you still report the mean and std of other experiments from Table 2?
> >
> > W4: You wrote “our method approximates arbitrary convolutions and fully connected layers”, however, in your reply to Q1 of reviewer zHBj you write “This decomposition is particularly effective for representing low-rank matrices”. So, the proposed method is not for arbitrary FC layers, especially if you fix m = 2. Is the method from [4] strictly less expressive than CDFlow layers for m = 2? Overall, I think that your paper would benefit from explicit comparison to this method.
> >
> > Q1: Thank you, I have no further questions, except for “What about using only one circulant matrix without diagonal ones?” already asked in "W1, NLL on synthetic task" above.
> >
> > Q2: What is p?
> >
> > Q3: What is the standard deviation of the times you mentioned for Emerging and for your method?

---

> > > ### Author Response · Authors · 2025-08-05
> > >
> > > We thank the reviewer for the thoughtful questions and constructive feedback, which helped improve the clarity and completeness of our work. Below, we respond in detail.
> > >
> > > >W1: NLL on synthetic task
> > >
> > > Thanks for noticing that circulant result on synthetic results is inspiring, which demonstrates the value of learning circulant structure in deep learning. It should be noted that circulant matrix alone is a special case of our method. Besides, a single circulant matrix suffers from following limitations: (1) its fixed structure cannot approximate arbitrary matrices, unlike our more flexible parameterization; (2) it has a limited parameter count and cannot adapt to varying task complexity, while our method allows tuning via the decomposition factor m; (3) its effectiveness is demonstrated only on synthetic data and may not generalize well on complex datasets(Table 7).
> > >
> > > Table 7: BPD of circulant vs. DCD on Galaxy and CIFAR-10.
> > >
> > > |Dataset|Model|BPD|
> > > |-|-|-|
> > > |Galaxy|circ|1.99±0.005|
> > > ||ours|1.92±0.002|
> > > |CIFAR-10|circu|3.48±0.003|
> > > ||ours|3.31±0.004|
> > >
> > >
> > > >W2: Lack of in-depth analysis
> > >
> > > Our model comes from [6], which uses a 5-layer MLP on 2D toy datasets. In response to the suggestion for deeper comparison, we conducted additional experiments on Chessboard, Moons, and Circles datasets, comparing CDMLP with MLP and structured variants: BFMLP (ButterflyLayer), EMLP (EmergingLayer), and WMLP (WoodburyLayer). Our method consistently achieves competitive or better MSE, FID, and NLL, with far fewer parameters(Table 8). This validates its parameter efficiency and addresses the concern.
> > >
> > > Table 8: Comparison of flow matching models on 2D toy datasets.
> > >
> > > |Dataset|Model|Params|MSE↓|FID↓|NLL↓|
> > > |-|--|-|-|-|-|
> > > |Chessboard|CDMLP|**8453**|3.810|**0.0115**|**7.0425**|
> > > | |MLP|791042|3.796|0.0062|10.6853|
> > > | |BFMLP|26114|**3.789**|0.0125|8.2088|
> > > | |EMLP|1577474|3.863|0.0247|1.76e+22|
> > > | |WMLP|40514|3.761|0.0042|9.0551|
> > > |Moons|CDMLP|**8453**|**1.842**|**0.0016**|31.0329|
> > > | |MLP|791042|1.854|0.0022|24.1885|
> > > | |BFMLP|26114|1.886|0.0031|**17.9168**|
> > > | |EMLP|1577474|1.855|0.0023|36.9462|
> > > | |WMLP|40514|1.905|0.0016|47.6328|
> > > |Circles|CDMLP|**8453**|2.482|0.0069|**15.4922**|
> > > | |MLP|791042|**2.433**|**0.0019**|42.4687|
> > > | |BFMLP|26114|2.470|0.0032|21.6716|
> > > | |EMLP|1577474|2.517|0.0032|20.5252|
> > > | |WMLP|40514|2.547|0.0021|26.6795|
> > >
> > >
> > > >W3:  Mean and std of experiments in Table 2
> > >
> > > Thanks a lot for your clarification! The mean and std in Supplementary Table 2 offer a solid basis for assessing the stability of ours against others. We will strive to provide std values for all experiments; however, due to the large volume, this may not be completed within the rebuttal period. According to [7], Glow has 3.36±0.002 on CIFAR-10; ours is 3.31±0.004. In general, we believe normalizing flows tend to have small std values. We appreciate  reviewer's rigorous evaluation on experiment results, but we would like to highlight that our method achieves stable and competitive performance.
> > >
> > > >W4: Is the method from [4] strictly less expressive……
> > >
> > > The conv coupling layer in [4] replaces the affine coupling layer in standard multi-scale flow, resulting in totaling ~5.9M parameters.  In contrast, our coupling layer uses a 3-layer CNN with ~4.7M parameters.
> > >
> > > Despite with fewer parameters, ours performs better: [4] reports BPD 3.34 on CIFAR-10, while ours achieves 3.31, showing a more efficient and effective parameterization.
> > >
> > > Please note that GitHub repo for [4] is empty,  but we are still trying to reproduce.
> > >
> > > >Q2: What is p?
> > >
> > > The parameter $p$ defines block size for partitioning a non-square matrix into square circulant blocks[5].  Note that experiment model is often with square cases.
> > >
> > > >Q3: Std of log-det time
> > >
> > > We report mean ± std of log-determinant computation time (20 runs, batch 128, input 32×32). Ours consistently achieves lower runtimes and variances across all channel sizes.
> > >
> > > Table 3. Log-det time (ms)
> > >
> > > |Channel|Ours|Emerging|
> > > |-|-|-|
> > > |16|0.109±0.008|0.125±0.010|
> > > |32|0.109±0.010|0.122±0.012|
> > > |48|0.110±0.008|0.122±0.010|
> > > |64|0.109±0.009|0.124±0.010|
> > > |80|0.111±0.009|0.124±0.012|
> > > |96|0.109±0.009|0.124±0.015|
> > > |128|0.109±0.010|0.123±0.012|
> > > |256|0.112±0.009|0.125±0.012|
> > >
> > > We are doing further experiments to add mean and std of other methods for comparison. Given existing experimental results and analysis of ours, we believe responses above has addressed reviewer's major concerns, both conceptually and empirically, and we sincerely hope our responses have contributed to a positive evaluation.
> > >
> > > [6]Lipman, Yaron, et al. "Flow matching for generative modeling." arXiv preprint arXiv:2210.02747 (2022).
> > >
> > > [7]Hoogeboom, Emiel, et al. "Emerging convolutions for generative normalizing flows."ICML'22

---

> > > > ### Comment · Reviewer_bzQ3 · 2025-08-06
> > > >
> > > > W2: I have reread the conversation and noticed that I missed both the fact that you trained a flow matching model, as I initially asked in this point, and that you have already reported metrics for MLP in your first response. Sorry for that! Anyway, the new table is much more informative. Do you know what happened to EMLP on Chessboard?
> > > >
> > > > Overall, the work you did for rebuttal is impressive, and you did address the first three weaknesses from my initial review. Thus, I have no choice but to raise my score to 4. However, to be completely honest, I am still disappointed by the fact that the paper does not suggest some in-depth insight about why the proposed parameterization is better than the others.

---

> > > > > ### Author Response · Authors · 2025-08-07
> > > > >
> > > > > We sincerely thank the reviewer for the careful reading, constructive feedback, and for increasing the score—your recognition is truly encouraging and motivates us to further improve our work.
> > > > >
> > > > > >W2:  what happened to EMLP on Chessboard
> > > > >
> > > > > In the initial training, EMLP was able to generate valid Chessboard images, and both MSE and FID appeared normal. As a result, we did not further investigate the abnormal NLL values. After your comment, we retrained EMLP on the Chessboard dataset and obtained: MSE = 3.797, FID = 0.0165, and NLL = 9.357. Compared to our CDMLP results (MSE = 3.810, FID = 0.0115, NLL = 7.0425), EMLP does not show a clear advantage.
> > > > >
> > > > > >in-depth insight about why the proposed parameterization is better.
> > > > >
> > > > > From a theoretical standpoint, our parameterization offers favorable trade-offs in both computation and memory. CDFlow requires $\mathcal{O}(mn)$ time for log-determinant computation and $\mathcal{O}(mn \log n)$ for inversion, where $m$ is a small decomposition factor (typically 2). In contrast, $1\times1$ Conv and Periodic layers require $\mathcal{O}(n^3)$, while ButterflyFlow requires $\mathcal{O}(n^2)$.
> > > > >
> > > > > In terms of memory, dense linear layers store $\mathcal{O}(n^2)$ parameters. ButterflyFlow reduces this to $\mathcal{O}(n \log n)$ through recursive structure, and our method further lowers it to $\mathcal{O}(mn)$, providing a more compact yet expressive alternative.
> > > > >
> > > > > Regarding expressiveness, we find no widely accepted framework to rigorously compare the representational capacity of different structured transformations in the context of normalizing flows. Existing methods rarely analyze or quantify their effect on flow expressiveness, making theoretical comparisons challenging.
> > > > >
> > > > > Instead, we present extensive empirical evidence—including BPD, parameter counts, and Flow Matching evaluations—demonstrating that our method consistently delivers competitive or superior performance with lower complexity, validating the practical benefits of our approach.
> > > > >
> > > > > We will strive to reflect these points in the camera-ready version and hope this helps address the reviewer’s concerns.

---

### Official Review · Reviewer_zHBj · 2025-07-03

**Clarity:** 3
**Significance:** 2
**Originality:** 2
**Rating:** 5
**Confidence:** 3

**Summary:**

This paper is concerned with the efficient implementation of normalizing flows for sampling from complex probability distributions. These models require the construction of invertible deep networks, with the extra condition that the determinant of the Jacobian matrix should be easy to compute. The inverse is important primarily for the sampling process of the model, while the log-det is important primarily for the training process. Based on this, the authors focus on the efficient implementation of linear transformations (that is, linear layers) whose determinants and inverses are easy to compute. They leverage a result on the approximation of dense matrices by the product of structured matrices: namely the alternating product of diagonal and circulant matrices. The diagonal matrices have easily computable inverses and determinants, and the circulant matrices can be similarly handled using FFTs.

**Questions:**

My questions are primarily around the theoretical properties of this approach as they pertain to training normalizing flow models.

As noted in the weaknesses, there is not much discussion on how the choice of $m$ affects the expressivity of the model. Of course, as $m$ increases the linear layer can parameterize a larger family of matrices. When $m$ is small, what kind of matrices are expected to be well-approximated? For instance, the composition of convolutions and pointwise multiplications might do a good job approximating “spatially-varying convolutions,” pseudodifferential operators, or something similar.

This method is put forth with the idea of enabling efficient matrix inversion and log-determinant calculation. However, these fail when the component (diagonal/circulant) matrices are not well-conditioned. Did you employ regularization in training to ensure that this does not happen? Is it even worthwhile to add penalties to enforce well-conditioned linear layers?

You have elected to focus on products of diagonal and circulant matrices, particularly since the latter can be easily parameterized using FFTs. However, there are many other types of structured matrices that admit similar efficient $n\log n$ computations; see the paper “Butterfly factorization” by Li et. al. Do the algorithms and approximations in that paper have any bearing on potential generalizations of your proposed method? A *thorough* discussion on this would make this paper much more significant, even if not much is done experimentally.

There is a claim on limitations that I do not understand. On pg. 6, you say that “A current limitation of [your] approach is its reliance on 1x1 convolutions.” But your convolutions are implemented directly in the Fourier domain, so it does not seem that you have any control over the spatial extent of the convolutions implemented by the circulant matrices. What did you mean by this statement?

**Ethical Concerns:**

["NO or VERY MINOR ethics concerns only"]

**Final Justification:**

Thank you for your rebuttal, and for addressing most of my comments – I hope you incorporate these remarks into the camera-ready version, if accepted. There is a lingering issue with the discussion around 1x1 convolutions. To my understanding, “1x1 convolutions” when represented as circulant matrices are just constant multiples of the identity matrix (in the spatial domain). When there are multiple channels, perhaps there is mixing of channels, but the point still stands. If you parameterize the convolution as a Fourier multiplier, which you do in this work, you lose control over the spatial extent of the convolution filter, so the convolutions are no longer 1x1. Your comment on the transformation being square-shaped doesn’t seem particularly relevant to this. I ask that you please think about this a bit more and make the final version of the paper more clear on this matter.

Based on your response, I am increasing my rating to 5 (accept).

**Limitations:**

Yes

**Paper Formatting Concerns:**

Readability of some figures was not so great. It would be better to use fewer (but larger) images in Figure 2a, 2b, and perhaps put more examples in an appendix. Additionally, Figure 3 is illegible when not viewed on a color screen (e.g., printed in black and white). Please put some more effort into making the plot’s results more clear in such contexts.

**Quality:**

3

**Strengths And Weaknesses:**

Strenghths:

The results and methods put forth in this work have the strength of being quite simple and easy to understand. The work leverages established results on structured matrix approximation to good effect, yielding a method that I can see being easily adapted by practitioners. The methods based on these results attain good asymptotic complexity, only comparable to the Woodbury method of [Lu and Huang, 2020], which is made up for with superior performance on test datasets and in runtime.

Weaknesses:

There are some missing references on prior works that use circulant-diagonal linear layers in machine learning applications. For instance, see “ACDC: A structured efficient linear layer” by Moczulski et. al. Papers that use such methods generally weren’t using circulant-diagonal linear layers to implement normalizing flows, so it is not necessary to “compare” as far as experiments go, but it would be best to acknowledge them. Additionally, you set $m=2$ in your experiments, which is presumably quite a bit less than the $m=n$ necessary for exact representation of any dense matrix. Although it is acknowledged that “the approximation error can be controlled by adjusting the number of factors,” not much else is said on the expressivity of approximation as a function of $m$.

---

> ### Author Rebuttal · Authors · 2025-07-31
>
> We sincerely thank the reviewer for the thoughtful feedback and valuable suggestions, which greatly helped refine both the theoretical foundation and clarity of our work. We address each point below.
> >Weakness1：There are some missing references on prior works……
>
> Thank you for pointing this out. We will include a citation to “ACDC: A Structured Efficient Linear Layer” (Moczulski et al.) to acknowledge prior usage of circulant-diagonal layers. While they do not compute the inverse transformations or the log-determinant of the Jacobian, both approaches share the goal of efficient linear transformations.
>
> >Weakness 2 & Question 1：As noted in the weaknesses, there is not much discussion on how the choice of affects the expressivity of the model.
>
> In practice, we aim to balance expressivity with efficiency in parameter count and runtime. To support the selection of the hyperparameter $m$ (i.e., $m$ diagonal and $m - 1$ circulant) , we conducted additional experiments.
>
> First, we replicated standard experiments on the expressivity of structured matrices following the setup in [2]. We added comparisons using circulant and DCD structures. As shown in Table 1, circulant layers achieve fast convergence and closely approximate full matrices. Notably, even with $m=2$, the DCD structure significantly outperforms LU and its components, demonstrating strong approximation capabilities with minimal complexity.
>
> Table 1. Mean NLL ± 95% CI on synthetic data (step 5000 / 10000)
> | Type| 5000 NLL |10000 NLL|
> | ----- | --- | -- |
> | full  | 49.11 ± 0.01 | 48.32 ± 0.12 |
> | tril  | 51.91 ± 0.34 | 50.89 ± 0.18 |
> | triu  | 50.27 ± 0.38 | 49.61 ± 0.73 |
> | trilu | 51.85 ± 0.49 | 51.04 ± 0.22 |
> | circ  | 49.07 ± 0.24 | 48.49 ± 0.39 |
> | dcd   | 50.18 ± 0.38 | 49.21 ± 0.12 |
>
> We then evaluated the relationship between $m$ and model expressivity, as well as $m$ and runtime performance.
> As shown in Table 2, increasing $m$ on CIFAR-10 yields only marginal improvements in model performance, while incurring noticeable increases in both parameter size and computation. We also measured forward and backward runtime across different values of $m$, summarized in Table 3 and Table 4, where we observed that runtime increases with $m$, yet still remains faster than existing baselines reported in the original paper.
>
> Table 2. Performance and parameter statistics of CDFlow under different values of $m$ on CIFAR-10.
> | Model | Setting | BPD (CIFAR-10) | #Params | DCD Params |
> |--------|---------|----|---------|------------|
> | CDFlow| m=2 | 3.31  | 44.2 M  | 6.9 K  |
> | CDFlow| m=3 | 3.30  | 44.2 M  | 11.1 K |
> | CDFlow| m=4 | 3.30  | 44.2 M  | 15.3 K |
>
> Table 3. Forward computation time (ms) for CD-Linear layers with different values of m, compared against traditional  convolutions.
>
> | Channels | $m=2$ | $m=3$ | $m=4$ | $m=5$ | $m=6$ |1x1 Conv|
> |---|---|---|---|--|--|---|
> |16 |0.226|0.226|0.326|0.374 |0.424 |0.240|
> |32|0.323|0.405 | 0.547|0.690 |0.837 |0.432 |
> |48|0.379|0.575 | 0.792|1.008 |1.225 |0.477 |
> |64|0.454|0.729 | 1.007|1.287 |1.571 |0.535 |
> |80|0.557|0.909 | 1.267|1.625 |1.983 |0.679|
> |96|0.719|1.194 | 1.676| 2.169 |2.644 |0.758|
> |128 | 0.875|1.434|1.963|2.529|3.144|0.880|
>
> Table 4. Inverse computation time (ms) for CDLinear layers with different $m$ values, compared against $1 \times 1$ convolutions.
>
> | Channels | $m=2$ | $m=3$ | $m=4$ | $m=5$ | $m=6$ |1x1 Conv |
> |---|---|--|---|---|---|---|
> | 16 | 0.170 | 0.229 | 0.291 | 0.353 | 0.416  | 0.195 |
> | 32 | 0.268 | 0.407 | 0.549 | 0.697 | 0.836  | 0.328 |
> | 48 | 0.348 | 0.566 | 0.782 | 0.993 | 1.234  | 0.428 |
> | 64 | 0.428 | 0.706 | 1.011 | 1.264 | 1.544  | 0.511 |
> | 80 | 0.532 | 0.890 | 1.275 | 1.598 | 1.972  | 0.714 |
> | 96 | 0.693 | 1.168 | 1.653 | 2.135 | 2.633  | 0.757 |
> | 128| 0.849 | 1.397 | 1.940 | 2.492 | 3.054  | 0.935 |
>
> Additionally, we tested log-determinant computation time as the channel dimension varies from 16 to 1024. As shown, our method maintains stable performance across different $m$, with runtime ranging narrowly from 0.095 ms to 0.097 ms, demonstrating robustness to input dimensionality.
>
> Considering the trade-off between expressivity and efficiency, we adopt $m = 2$ as the default configuration throughout our experiments unless otherwise specified.
>
> >Question 1: what kind of matrices are expected to be well-approximated?
>
> This decomposition is particularly effective for representing low-rank matrices. Prior work [1] shows that any $n \times n$ matrix of rank $k$ can be approximated using a product of $k$ diagonal and $k{-}1$ circulant matrices. Moreover, networks built with such decompositions exhibit strong expressivity: [1] proves that any ReLU network of width $n$ and depth $L$ can be approximated by a diagonal-circulant neural network (DCNN) of depth $(2n{-}1)L$, and DCNNs can represent functions that are not expressible by any rank-$k$ ReLU network, highlighting their superior modeling capacity.
>
> Our method extends this idea by expressing convolutional operations as matrix multiplications and applying structured decomposition to the resulting weight matrices. As a result, any convolution operation that can be formulated via matrix multiplication—including discrete convolutions and their linearized forms—can in principle be approximated within our framework. In particular, circular convolutions are naturally suited, as their Jacobians correspond to circulant matrices.
>
> >Question 2:these fail when the component (diagonal/circulant) matrices are not well-conditioned.
>
> As discussed in [3], randomly initialized circulant matrices have stable minimum singular values and low condition numbers. Compared to Toeplitz or Gaussian matrices, circulants offer better numerical stability.
> Practically, we carefully design the initialization of both diagonal and circulant components. The diagonal matrices are sampled from a normal distribution centered at 1 with variance scaling inversely with channel size, i.e., $\mathcal{N}(1, 1/c^2)$, to avoid vanishing or exploding scales. The circulant components are initialized in the frequency domain from $\mathcal{N}(1, I)$, leveraging the orthogonality of the DFT basis. These schemes result in well-conditioned composite matrices with condition numbers typically close to 1 at the start of training.
>
> In addition, we apply spectral normalization (Lines 233–234 in the paper) to further stabilize training. This constrains the spectral norm of each weight matrix $W$ to 1, effectively limiting the network’s Lipschitz constant. Since circulant matrices are diagonalizable via the discrete Fourier transform (DFT), their singular values correspond to the magnitudes of their frequency-domain representations. Therefore, spectral normalization can be efficiently implemented by normalizing each frequency-domain vector by its maximum modulus, ensuring both robustness and efficiency.
>
> >Question 3: However, there are many other types of structured matrices that admit similar efficient……
>
> We thank the reviewer for the insightful suggestion and for pointing us to the Butterfly factorization literature[4] The butterfly factorization approximates an $N \times N$ matrix $K$ satisfying the complementary low-rank property by a product of sparse matrices. This enables matrix--vector multiplication with $\mathcal{O}(N \log N)$ complexity and $\mathcal{O}(N \log N)$ parameters.
>
> In comparison, our diagonal-circulant decomposition requires only $\mathcal{O}(MN)$ parameters, where $M$ is a small constant. More importantly, one of the major advantages of our method lies in the efficient implementation of matrix inverse and log-determinant computations---both critical operations in flow-based generative models. As shown in Table 1 of the paper, our inverse computation has $\mathcal{O}(n \log n)$ complexity due to the structured FFT-based approach, whereas the inverse of butterfly-structured matrices typically incurs $\mathcal{O}(n^2)$ cost.
>
> >Question 4: There is a claim on limitations that I do not understand……
>
> Lastly, we would like to clarify the current limitations of our method. As discussed earlier, our approach is capable of approximating any convolution operation that can be expressed as a matrix multiplication. However, in the context of flow-based models, it is essential to compute both the inverse of the weight matrix and the corresponding Jacobian determinant. These operations are generally only well-defined for square matrices, which constrains our method's applicability to $1 \times 1$ convolutions or other square-shaped transformations. That said, our ongoing work focuses on extending the framework to support determinant and inverse computations for non-square matrices. We believe this will enable the application of our method to general $d \times d$ convolutions and potentially to a wider range of structured transformations.
>
> We sincerely thank the reviewer for the thoughtful feedback and insightful questions. We hope our responses have addressed the concerns regarding model expressivity, training stability, and potential generalizations. We will also revise the final version of the paper to include additional references, including prior works such as ACDC and Butterfly Factorization, to better situate our contribution within the broader literature.
>
> [1] Araujo, Alexandre, et al. "Understanding and training deep diagonal circulant neural networks." ECAI 2020. IOS Press, 2020. 945-952.
>
> [2] Chao, Chen-Hao, et al. "Training energy-based normalizing flow with score-matching objectives." Advances in Neural Information Processing Systems 36 (2023): 43826-43851.
>
> [3] Pan, Victor Y., and Guoliang Qian. "Condition numbers of random toeplitz and circulant matrices." arXiv preprint arXiv:1212.4551 (2012).
>
> [4]Li, Yingzhou, et al. "Butterfly factorization." Multiscale Modeling & Simulation 13.2 (2015): 714-732.
>
> [5]Dao, Tri, et al. "Learning fast algorithms for linear transforms using butterfly factorizations." International conference on machine learning. PMLR, 2019.

---

> > ### Comment · Reviewer_zHBj · 2025-08-05
> >
> > Thank you for your rebuttal, and for addressing most of my comments – I hope you incorporate these remarks into the camera-ready version, if accepted. There is a lingering issue with the discussion around 1x1 convolutions. To my understanding, “1x1 convolutions” when represented as circulant matrices are just constant multiples of the identity matrix (in the spatial domain). When there are multiple channels, perhaps there is mixing of channels, but the point still stands. If you parameterize the convolution as a Fourier multiplier, which you do in this work, you lose control over the spatial extent of the convolution filter, so the convolutions are no longer 1x1. Your comment on the transformation being square-shaped doesn’t seem particularly relevant to this. I ask that you please think about this a bit more and make the final version of the paper more clear on this matter.
> >
> > Based on your response, I am increasing my rating to 5 (accept).

---

### Author Response · Authors · 2025-08-09
**Appreciation to Reviewers**

We are deeply grateful to all reviewers for your time, thoughtful feedback, and constructive suggestions—whether identifying issues, recommending relevant literature, or encouraging additional experiments—which have significantly enhanced the technical quality, clarity, and completeness of this work. We also value the recognition and encouragement reflected in your review scores, which motivate us to further advance this research, and we look forward to building upon these insights in our future endeavors.

---

### Decision · Program_Chairs · 2025-09-17

**Decision:**

Accept (poster)

**Comment:**

This paper proposes CDFlow, which replaces dense 1×1 mixing layers with a product of diagonal and circulant matrices to enable efficient computation of invertible layers in normalizing flows.

The paper received generally positive reviews: one borderline accept (4) and three accepts (5). The main concerns raised during the review and rebuttal phases were:

- **Technical novelty**: While circulant–diagonal networks have been studied previously, reviewers questioned whether the contribution is incremental.
- **Missing related work**: Several relevant papers (e.g., ACDC) were initially omitted.
- **Clarity issues**: Reviewers highlighted unclear discussions on the reliance on 1×1 convolutions versus Fourier-domain mixing, inconsistencies in complexity statements, vague use of terms such as “compact representation,” and imprecise explanations of determinant computation.
- **Experiments**: The experimental section lacked error bars and a deeper parameter–quality–runtime trade-off analysis. Figures were noted to be small and somewhat unclear.

The author rebuttal was thorough and convincing. Reviewers acknowledged that most concerns were resolved in substance, with remaining issues confined to clarity and presentations in experiment ****that can be fixed in the camera-ready version. After carefully reading the reviews and rebuttal discussions, AC agrees with the reviewers’ decision and recommends acceptance.